# An in situ dual-anchoring strategy for enhanced immobilization of PD-L1 to treat autoimmune diseases

Shenqiang Wang [1], Ying Zhang[1], Yanfang Wang[1], Yinxian Yang[1], Sheng Zhao[1], Tao Sheng[1], Yuqi Zhang [1,2,3], Zhen Gu [1,3,4,5,6,7] ✉, Jinqiang Wang [1,3,6,8] ✉ & Jicheng Yu [1,3,4,5,6] ✉

Immune checkpoints play key roles in maintaining self-tolerance. Targeted potentiation of the checkpoint molecule PD-L1 through in situ manipulation offers clinical promise for patients with autoimmune diseases. However, the therapeutic effects of these approaches are often compromised by limited specificity and inadequate expression. Here, we report a two-step dual-anchor coupling strategy for enhanced immobilization of PD-L1 on target endogenous cells by integrating bioorthogonal chemistry and physical insertion of the cell membrane. In both type 1 diabetes and rheumatoid arthritis mouse models, we demonstrate that this approach leads to elevated and sustained conjugation of PD-L1 on target cells, resulting in significant suppression of autoreactive immune cell activation, recruitment of regulatory T cells, and systematic reshaping of the immune environment. Furthermore, it restores glucose homeostasis in type 1 diabetic mice for over 100 days. This specific in situ bioengineering approach potentiates the functions of PD-L1 and represents its translational potential.

Autoimmune diseases, such as type 1 diabetes (T1D), rheumatoid arthritis (RA), and systemic lupus erythematosus developed from the imbalance of the immune systems, have become the third most common category of diseases[1]. In the progression of autoimmune diseases, T cells, the critical orchestrator of autoimmunity, could be activated by autoantigens, thus prone to attack peripheral healthy tissues[2,3]. Therefore, suppressing immune-inflammatory activity has been the primary therapeutic approach for treating autoimmune diseases[4,5]. Even though immunosuppressive drugs or antibodies could partially inhibit autoreactive immune cell functions, the various

side effects and insufficiently effective medication have severely reduced the quality of life for patients and increased their economical burdens[6,7]. Thus, the development of immunological strategies to specifically regulate the activity of tissue-infiltrating T cells and prevent autoimmune attacks has been a promising solution for rebalancing the immune system in the treatment of autoimmune diseases.

Programmed death 1 (PD-1), an essential immune checkpoint factor expressed by activated T cells and B cells, plays a critical role in regulating the self-tolerance of the immune system[8,9]. Preclinical investigations revealed that intact interaction between PD-1 and its

[1]Zhejiang Provincial Key Laboratory for Advanced Drug Delivery Systems, College of Pharmaceutical Sciences, Zhejiang University, Hangzhou 310058, China. [2]Department of Burns and Wound Center, Second Affiliated Hospital, School of Medicine, Zhejiang University, Hangzhou 310009, China. [3]National Key Laboratory of Advanced Drug Delivery and Release Systems, Zhejiang University, Hangzhou 310058, China. [4]Liangzhu Laboratory, Zhejiang University Medical Center, Hangzhou 311121, China. [5]Department of General Surgery, Sir Run Run Shaw Hospital, School of Medicine, Zhejiang University, Hangzhou 310016, China. [6]Jinhua Institute of Zhejiang University, Jinhua 321299, China. [7]MOE Key Laboratory of Macromolecular Synthesis and Functionalization, Department of Polymer Science and Engineering, Zhejiang University, Hangzhou 310027, China. [8]Department of Pharmacy, Second Affiliated Hospital, Zhejiang University School of Medicine, Zhejiang University, Hangzhou 310009, China. ✉e-mail: guzhen@zju.edu.cn; jinqiang_wang@zju.edu.cn; yujicheng@zju.edu.cn

ligand PD-L1 could inhibit T cell activation, leading to their exhaustion or apoptosis, thereby lessening the severity of numerous autoimmune diseases[10,11]. Recent studies elaborated that the administration of PD-L1-overexpressed exogenous cells or cell-derived extracellular vesicles could reduce infiltrating autoreactive T cells and promote the recruitment of infiltrating regulatory T cells (Tregs) to reverse early-onset hyperglycemia in nonobese diabetic (NOD) mice, prevent multiple sclerosis in autoimmune encephalomyelitis, and ameliorate ulcerative colitis and psoriasis[12–16]. However, the occurrence of

detrimental pulmonary embolism and the undesired systemic immunosuppression restrict their clinical applications[17].

Herein, we develop a dual-anchor coupling strategy to in situ bioengineer endogenous target cells for enhanced bioorthogonal immobilization of PD-L1 on the cell surface (Fig. 1a). Briefly, reactive oxygen species (ROS)-responsive nanoparticles are first designed to deliver azido-conjugated unnatural sugar (Ac4ManNAz) to the inflammatory target tissue, and then the target cells express azido groups on the surface via metabolic glycoengineering. Subsequently, the

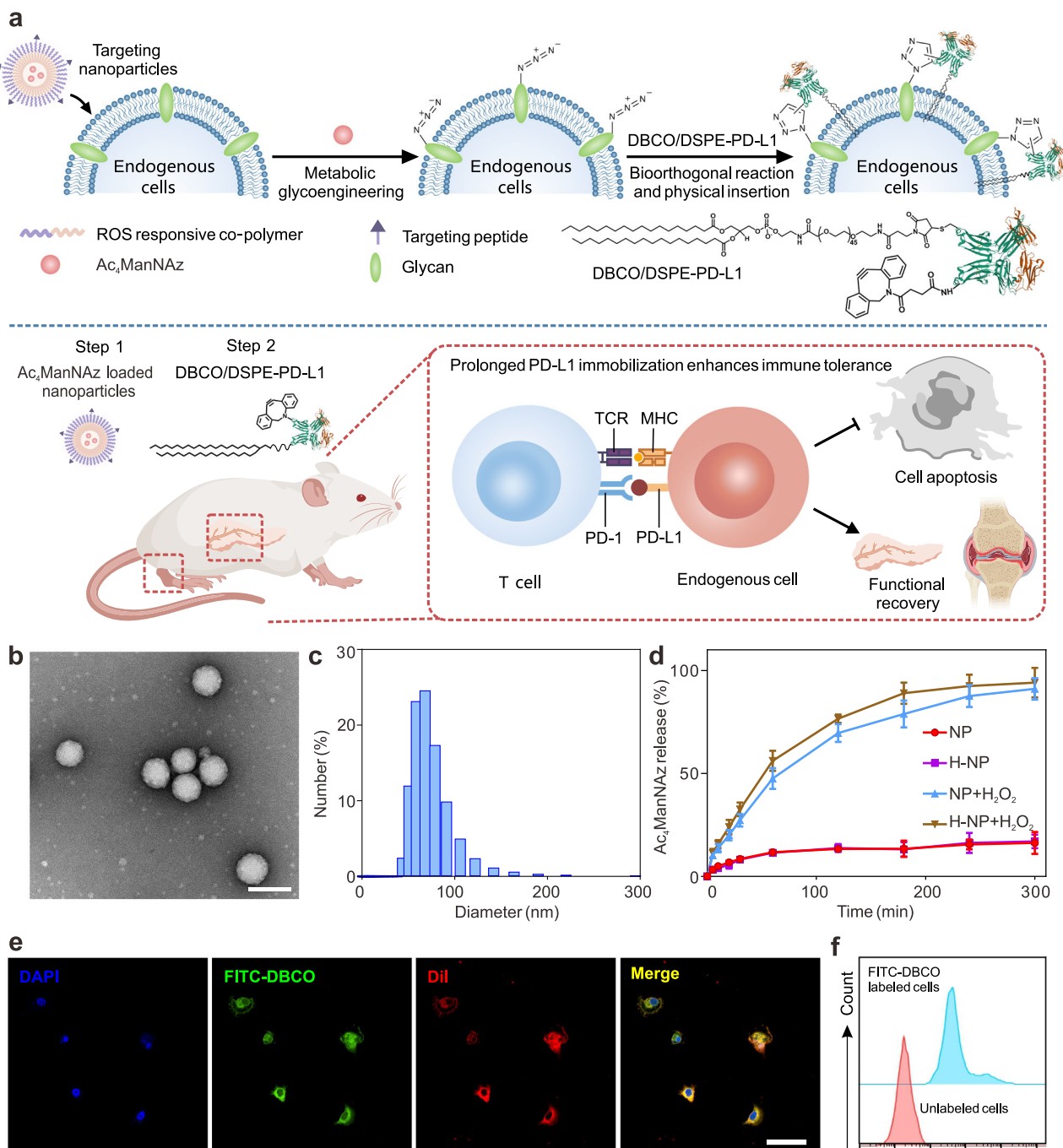

**Fig. 1 | Prolonged immobilization of immune checkpoint for the treatment of autoimmune diseases. a** Scheme of the immobilization of PD-L1 molecules on targeted cell membrane to enhance immune tolerance for the treatment of T1D and RA. **b** TEM image of H-NPs. Scale bar: 100 nm. **c** The diameter of H-NPs was determined by dynamic light scattering (DLS). **d** The ROS-responsive release profiles of Ac4ManNAz. NPs represent MPEG5k-P(DMAEMA-PBA)14k nanoparticles. The

concentration of H2O2 was 1 mM. Data represent the mean ± s.d. (n = 3 independent samples). **e** Confocal imaging of Min 6 cells incubated with FITC-DBCO after a 3-day introduction of Ac4ManNAz. Scale bar: 50 μm. **f** Flow cytometry of unlabeled Min 6 cells and FITC-labeled Min 6 cells through metabolic glycoengineering and click chemistry.

systemically administered dibenzocyclooctyne (DBCO) and amphiphilic lipid (1,2-distearoyl-sn-glycero-3-phosphoethanolamine, DSPE) co-functionalized PD-L1 (DBCO/DSPE-PD-L1) could be immobilized onto the surface of cells by the azide-alkyne "click" reaction together with the insertion of lipid tails into the cell membrane. Although in situ foreign protein engineering on endogenous cells using bioorthogonal reactions has been reported, its clinical applications have been limited due to the short modulation duration caused by glycan/membrane recycling[18]. Our double-anchor coupling strategy based on the integration of bioorthogonal chemistry and physical insertion is able to prevent the undesired loss of the conjugated proteins caused by the metabolic activity of cells, thus significantly enhancing the retention of the PD-L1 proteins on the target cells. Animal studies further demonstrated that the bioengineered endogenous cells reversed early-onset type 1 diabetes (T1D) and remitted the development of rheumatoid arthritis (RA) by improving immune tolerance and diminishing the attack from autoreactive T cells. Moreover, this approach could eliminate the potential nonspecific immune responses and immune-related toxicity, which may be applicable for the treatment of autoimmune diseases.

## Results

### Preparation of ROS-responsive nanoparticles

The development of autoimmune disease is often accompanied by the presence of inflammation, resulting in mass production of reactive oxygen species (ROS)[19,20]. Therefore, a ROS-responsive co-polymer polyethylene glycolyl monomethyl ether-poly((2-dimethylamino)ethyl methacrylate-4-(bromomethyl)phenylboronic acid) (MPEG$_{5k}$-P(DMAEMA-PBA)$_{14k}$) was synthesized according to our previous report (Supplementary Figs. 1 and 2)[21]. Afterward, a hybrid nanoparticle (H-NP) was prepared from the resulting ROS-responsive co-polymer and N-hydroxysuccinimide-polyethylene glycolyl-poly(lactic-co-glycolic acid) (NHS-PEG$_{5k}$-PLGA$_{6k}$) using the nanoprecipitation method, which allows further modification of targeting molecules on the surface via the NHS groups. Poly(vinyl alcohol) (PVA) was incorporated to stabilize the nanoparticles through the formation of acid-inert ester bonds between PBA and cis −1, 3-diols on PVA. The resulting H-NPs have an average diameter of around 77.8 nm (Fig. 1b, c). Notably, the H-NPs exhibited a high loading capacity (19.1%) for Ac$_4$ManNAz, perhaps due to hydrophobic-hydrophobic and electrostatic interactions (Supplementary Fig. 3).

In the presence of hydrogen peroxide (H$_2$O$_2$), the PBA groups of MPEG$_{5k}$-P(DMAEMA-PBA)$_{14k}$ could be oxidized and hydrolyzed, generating MPEG$_{5k}$-P(DMAEMA)$_{6k}$ with reduced stability and less positive charges (Supplementary Fig. 4a). During this process, H-NPs gradually disassociated, leading to the release of Ac$_4$ManNAz, as evidenced by changes in particle diameter and transmission electron microscope (TEM) images (Supplementary Fig. 4b, c). Under oxidative conditions, 88.9% of the encapsulated Ac$_4$ManNAz was released within 3 h (Fig. 1d). In addition, both Ac$_4$ManNAz and H-NPs showed negligible cytotoxicity (Supplementary Fig. 5).

### Prolonged expression of PD-L1 on target cells inhibited immune cells activation in vitro

Metabolic glycoengineering was utilized to decorate pancreatic cells with PD-L1, since metabolic glycoengineering enables the facile chemical decoration of biomacromolecules on target cells by manipulating cellular metabolisms to regulate glycosylation without complex genetic engineering techniques[22–24]. By incubating insulin-producing insulinomas Min 6 cells with Ac$_4$ManNAz for 3 days, the cells expressed azido groups on the cell membrane protein through the conversion of intracellular ManNAz into an azido sialic acid derivative (Fig. 1e, f). However, the duration of cell decoration was significantly limited due to glycan/membrane recycling and mitotic division[13].

Physical insertion by anchoring amphiphilic lipids into the cell membrane possesses high cell labeling property, which has been restricted by poor specificity[25–28]. Therefore, we proposed a dual-anchor coupling strategy by taking advantage of the two mentioned cell labeling approaches. Dibenzocyclooctyne (DBCO) functionalized NHS ester was firstly conjugated to PD-L1, resulting in DBCO-modified PD-L1 (DBCO-PD-L1) through a typical amine-NHS coupling reaction. Then, maleimide-functionalized DSPE-PEG (DSPE-PEG$_{2k}$-Mal) was conjugated to the DBCO-PD-L1 via a Mal-thiol reaction, yielding a bifunctionalized PD-L1 analog (DBCO/DSPE-PD-L1) (Fig. 2a). UV-visible spectrum identified that DBCO ligands could be conjugated to PD-L1 (Supplementary Fig. 6a)[18]. In addition, the resulting amphiphilic DBCO/DSPE-PD-L1 formed micelles with an average diameter of around 49.2 nm (Supplementary Fig. 6b).

To evaluate the PD-L1 engineering efficacy, the azido-expressed Min 6 cells were incubated with FITC-labeled DBCO-PD-L1 and DBCO/DSPE-PD-L1 for 1 h. Confocal microscopy and flow cytometry results showed that both of the PD-L1 analogs displayed the high cell conjugation efficiency, indicating that bioorthogonal chemistry strategy possessed a high binding performance (Supplementary Fig. 7a, b)[29–31]. Of note, after the prolonged incubation, the expression levels of DBCO-PD-L1-modified cells exhibited an obvious declination on day 3, whereas DBCO/DSPE-PD-L1-modified cells maintained a persistent expression of PD-L1 during the 14 days observation period (over 45%) (Fig. 2b–d, Supplementary Fig. 7c, Supplementary Fig. 8). The significantly prolonged expression behavior was attributed to the dual-anchor coupling approach, which effectively stabilized PD-L1 on the cell membrane. We further evaluated whether the structure of PD-L1 conjugates was appropriately folded by a fluorophore conjugated anti-PD-L1 and PD-1. More anti-PD-L1 and PD-1 were localized on DBCO/DSPE-PD-L1 labeled cell membrane compared to the cell alone group, indicating that the structure of PD-L1 conjugates was well maintained (Supplementary Fig. 9).

Restored expression of PD-L1 could enhance immune escape, as reported previously[32,33]. To investigate the immunoregulatory role of PD-L1-labeled cells on immune cells, CD3$^+$ T cells isolated from the lymph node of newly hyperglycemic NOD mice. Then, the activated T cells were incubated with PD-L1-labeled pancreatic cells (Fig. 2e). Confocal microscopy showed that T cells adhered closely to the DBCO/DSPE-PD-L1-labeled Min 6 cells. Nonetheless, such adhesion was attenuated in the unlabeled Min 6 cell group, suggesting that increased PD-L1 expression facilitated cell interaction through the PD-1/PD-L1 pathway (Fig. 2f). Furthermore, flow cytometric analysis was applied to assess the functional orientation of each cellular component in vitro. It was observed that anti-CD3/CD28 treatment induced interferon-γ (IFN-γ) and granzyme B (GzmB) production by CD8$^+$ T cells, while DBCO/DSPE-PD-L1-labeled cells remarkably reduced the frequency of IFN-γ and GzmB production (Supplementary Fig. 10, Fig. 2g, h). However, the introduction of anti-PD-L1 antibody or pretreated T cells with anti-PD-1 compromised the protective capacity of immobilized PD-L1 on the cell membrane, revealing that the interaction between PD-1 and PD-L1 contributed to the T cell exhaustion (Supplementary Fig. 11). We further verified that DBCO/DSPE-PD-L1-labeled cells increased FoxP3 expression as well as enhanced the population of Tregs, and the in vitro suppression assay confirmed the suppressive functionality of these CD4$^+$Foxp3$^+$ cells (Supplementary Fig. 12), which are crucial in suppressing immune response and maintaining self-tolerance[34].

### In situ immobilization of immune checkpoint reverses hyperglycemia in vivo

Type 1 diabetes mellitus, a T cell-mediated autoimmune disease characterized by the destruction of insulin-producing β cells and loss control of blood glucose, lacks effective treatment so far[35]. Encouraged by the in vitro results, we further assessed the performance of this dual-anchor coupling-based bioengineering approach in protecting

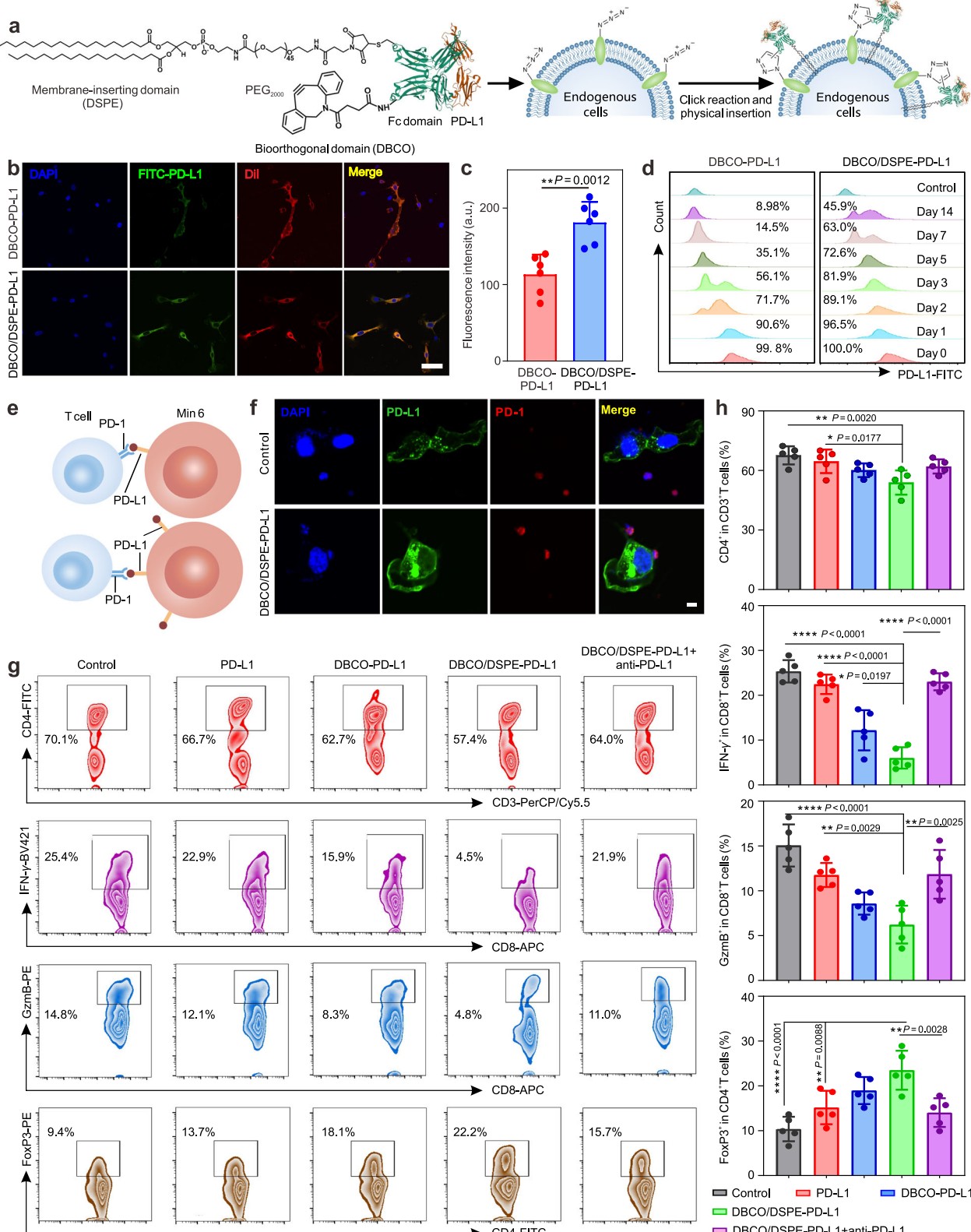

**Fig. 2 | Immobilization of PD-L1 on Min 6 cells inhibited the activation of immune cells in vitro. a** Schematic illustration indicated the modification of PD-L1 analog and the conjugation process of PD-L1 analog on Min 6 cells. **b**, **c** Confocal imaging (**b**) and quantitative assay (**c**) of Min 6 cells labeled with FITC-PD-L1 analogs on day 3. Scale bar: 50 μm. Data represent the mean ± s.d. ($n = 6$ independent samples). The data were analyzed by a two-tailed Student's $t$-test; $^{**}P < 0.01$. **d** Flow cytometry of Min 6 cells labeled with FITC-PD-L1 analogs at predetermined time points. **e** Schematic illustration revealed the interactions between unlabeled Min 6 cells or PD-L1-labeled Min 6 cells with lymph node T cells. **f** Colocalization of Min 6 cells or PD-L1-conjugated Min 6 cells and lymph node T cells were displayed by immunofluorescence staining. Scale bar: 5 μm. **g**, **h** The status (**g**) and quantitative analysis (**h**) of T cells in different treatment groups analyzed by flow cytometry. Data represent the mean ± s.d. ($n = 5$ biologically independent samples). The data were analyzed by one-way two-sided ANOVA; $^{****}P < 0.0001$, $^{**}P < 0.01$, $^{*}P < 0.05$.

pancreatic $\beta$ cells from immune attack and reversing early-onset hyperglycemia in vivo (Fig. 3a). We first evaluated the labeling efficiency of PD-L1 analogs on the endogenous $\beta$ cells. To selectively deliver Ac$_4$ManNAz to the pancreas, a ligand of the glucagon-like peptide-1 receptor (GLP1R) was conjugated to the H-NPs via a typical amine-NHS coupling reaction[36]. Through confocal microscopy imaging, we found that GLP1R peptide ligand conjugated H-NPs (GLP1R-H-NPs) were prone to be internalized into the Min 6 cells as compared to the free H-NPs (Supplementary Fig. 13). The in vivo biodistribution of H-NPs was then assessed by intravenous injection of Cy7.5-labeled H-NPs and GLP1R-H-NPs into NOD mice. GLP1R-H-NPs were detected in the pancreas 3 h post-administration, reaching a maximum at 6 h (Fig. 3b and Supplementary Fig. 14). The quantitative results further revealed that a dramatically higher fluorescence signal was found in the GLP1R-H-NP group compared to the non-targeting H-NP group, indicating effective delivery of cargo to the pancreas of NOD mice. Even though a high fluorescence in the liver of mice was observed, negligible toxicity or inflammation was found (Supplementary Fig. 15).

By targeted delivery of Ac$_4$ManNAz to the islet-rich pancreas, $\beta$ cells processed azido sialic acid derivatives on their cell membranes. Then, Cy5-labeled free PD-L1 and PD-L1 analogs (DBCO-PD-L1 and DBCO/DSPE-PD-L1) were intravenously administered after 3 days. Through fluorescent imaging, we confirmed distinct red signals appeared around the islets in the PD-L1 analog groups rather than in the free PD-L1 group, indicating that the bioorthogonal chemistry possessed high specificity with the targeted biomolecules (Supplementary Fig. 16a). Consistent with the in vitro results, an obvious signal attenuation was observed in the DBCO-PD-L1 group over time, which could be ascribed to glycan recycling (Fig. 3c and Supplementary Fig. 16b). In contrast, the persistent expression of PD-L1 in the DBCO/DSPE-PD-L1 group for over 21 days, indicating the dual-anchor coupling approach could significantly prolong the immobilization of immune checkpoints on target cells.

To further investigate the in vivo therapeutic efficiency, newly hyperglycemic NOD mice (blood glucose level > 250 mg dL$^{-1}$ for two consecutive days) were divided into four groups: untreated group, free PD-L1 group, DBCO-PD-L1 group, and DBCO/DSPE-PD-L1 group. The PD-L1 analogs groups were pretreated with the Ac$_4$ManNAz-loaded H-NPs. Blood glucose levels and body weight were monitored every two days until the endpoint (Fig. 3d, Supplementary Fig. 17). The blood glucose levels in the untreated mice increased and gradually reached severe hyperglycemia (blood glucose level > 600 mg dL$^{-1}$)[10,37]. Administration of free PD-L1 only slightly delayed the onset of diabetes but eventually led to hyperglycemia (Fig. 3e). In contrast, immobilization of DBCO-PD-L1 on the cell membrane through metabolic glycoengineering and click chemistry led to an improved control over blood glucose for a certain time. Strikingly, the progression of diabetes was significantly inhibited in NOD mice treated with DBCO/DSPE-PD-L1, and hyperglycemia was reversed to normoglycemia (71.4% on day 50). In the long-term evaluation, eight out of fourteen treated mice remained normoglycemia for over 100 days (Fig. 3f), and the survival of the NOD mice was prominently prolonged compared to other control groups (Fig. 3g). Through H&E staining, we confirmed that the introduction of DBCO/DSPE-PD-L1 preserved islets and reduced degree of insulitis (Fig. 3h and Supplementary Fig. 18a). Higher proportion of insulin$^+$ pancreatic $\beta$ cells and enhanced expression of PD-L1 in the pancreas revealed that the restored expression of PD-L1 could effectively prevent the loss of $\beta$ cells (Fig. 3i, j and Supplementary Fig. 18b, c).

To gain insights into the underlying mechanism, the status of pancreas-infiltrating T cells was investigated thoroughly. By collecting pancreas of the NOD mice, the phenotypes of immune cells were analyzed through immunofluorescence staining and flow cytometry, respectively. Less CD3$^+$ T cells (2.2 times reduction compared to the untreated group) were identified in the DBCO/DSPE-PD-L1 group on day 5 post-treatments, similar to the healthy mice group, indicating

that prolonged immobilization of PD-L1 efficiently inhibited T cell infiltration (Supplementary Fig. 19). With the treatment of PD-L1 analogs, the pancreas-infiltrating CD4$^+$ and CD8$^+$ T cells were significantly reduced (Fig. 4a–d and Supplementary Figs. 20 and 21). In particular, the mice treated with DBCO/DSPE-PD-L1 showed a 1.8 times reduction of the frequency of CD8$^+$ T cells compared to the untreated mice. Additionally, activated CD8$^+$ cytotoxic T cells could attack the pancreatic $\beta$-cells via secreting IFN-$\gamma$, GzmB, and perforin[12]. Therefore, the activation status of CD8$^+$ T cells was assessed. Over 12.5% and 11.8% of CD8$^+$ T cells in the pancreas of the untreated NOD mice were IFN-$\gamma$ and GzmB producing cytotoxic T lymphocytes, which resembled in the group treated with free PD-L1. In contrast, much decreased percentage of CD8$^+$IFN-$\gamma^+$ (5.0%) and CD8$^+$GzmB$^+$ (4.6%) T cells were detected in the DBCO/DSPE-PD-L1 group, suggesting that effector T cells were inhibited due to the prolonged immobilization of PD-L1 (Fig. 4e–h). The protective performance could become more pronounced after prolonged cultivation (Supplementary Fig. 22).

Moreover, PD-L1 plays a predominant role in facilitating the development and function of Tregs, which could improve immune tolerance[38]. Less Tregs were identified in the untreated group and free PD-L1 group, indicating that the development of T1D accelerated the loss of Tregs (Supplementary Figs. 23 and 24). Recent studies reported that the CD4$^+$CD49b$^+$ type 1 regulatory T (Tr1) cells could alleviate the progression of T1D[39]. Here, we validated that Tr1 cells were restored in the pancreas of NOD mice treated with PD-L1 analogs (Supplementary Fig. 25). Moreover, the pro-inflammatory cytokines including IFN-$\gamma$, tumor necrosis factor-$\alpha$ (TNF-$\alpha$), interleukin-1$\beta$ (IL-1$\beta$), and IL-6 also showed down regulation, whereas the level of anti-inflammatory cytokine (IL-10) was elevated in the DBCO/DSPE-PD-L1 group (Fig. 4i).

In order to explore whether the systemic and local compartments could be impacted, we collected the spleen and pancreatic lymph nodes and performed flow cytometry to evaluate the proportion of immune cells. With the introduction of DBCO/DSPE-PD-L1, the proportion of CD8$^+$ T cells in the pancreatic lymph nodes was significantly reduced compared to the other groups (Supplementary Figs. 26 and 27a, b). Previous report demonstrated that $\beta$-cell-specific CD8$^+$ T cells in NOD mice were primed in the pancreatic draining lymph node before they infiltrated the pancreas[3]. Thus, we believe that the onset of type 1 diabetes contributed to the elevation of CD8$^+$ cytotoxic T cells in the pancreatic lymph nodes, while the prolonged stabilization of PD-L1 on the pancreatic $\beta$ cells could not only reverse the infiltration of CD8$^+$ T cells, but restrain the activity of autoreactive T cells in the pancreatic lymph nodes. Moreover, the proportion of GzmB-positive CD8$^+$ T cells and IFN-$\gamma$-positive CD8$^+$ T cells were also reduced after the administration of DBCO/DSPE-PD-L1 (Supplementary Fig. 27c–f). The DBCO/DSPE-PD-L1 treated NOD mice enhanced the population of Tregs as evidenced by the increased expression of FoxP3 (Supplementary Fig. 27g, h). We further investigated the status of T cells in the spleen of NOD mice. Lower proportion of CD8$^+$ T cells was observed in the DBCO/DSPE-PD-L1-treated group compared to the untreated group and PD-L1-treated group (Supplementary Figs. 28 and 29). Insignificant difference was observed in regard to GzmB-positive CD8$^+$ T cells and IFN-$\gamma$-positive CD8$^+$ T cells, perhaps due to the fact that locally modulation of immune microenvironment could not significantly affect the systemic immune systems. Taken together, this dual-anchor coupling approach-based bioengineering of endogenous pancreatic $\beta$ cells could reshape the inflammatory microenvironment, protect pancreas from immune attack, and synergistically promote the reversion of early-onset T1D.

## In situ immobilization of immune checkpoint ameliorates collagen-induced arthritis (CIA)

In order to explore the potential of this PD-L1 dual-anchor coupling approach for the treatment of other autoimmune diseases, we further conducted a proof-of-concept evaluation in rheumatoid arthritis (RA). First, the in vitro cell labeling performance was investigated via

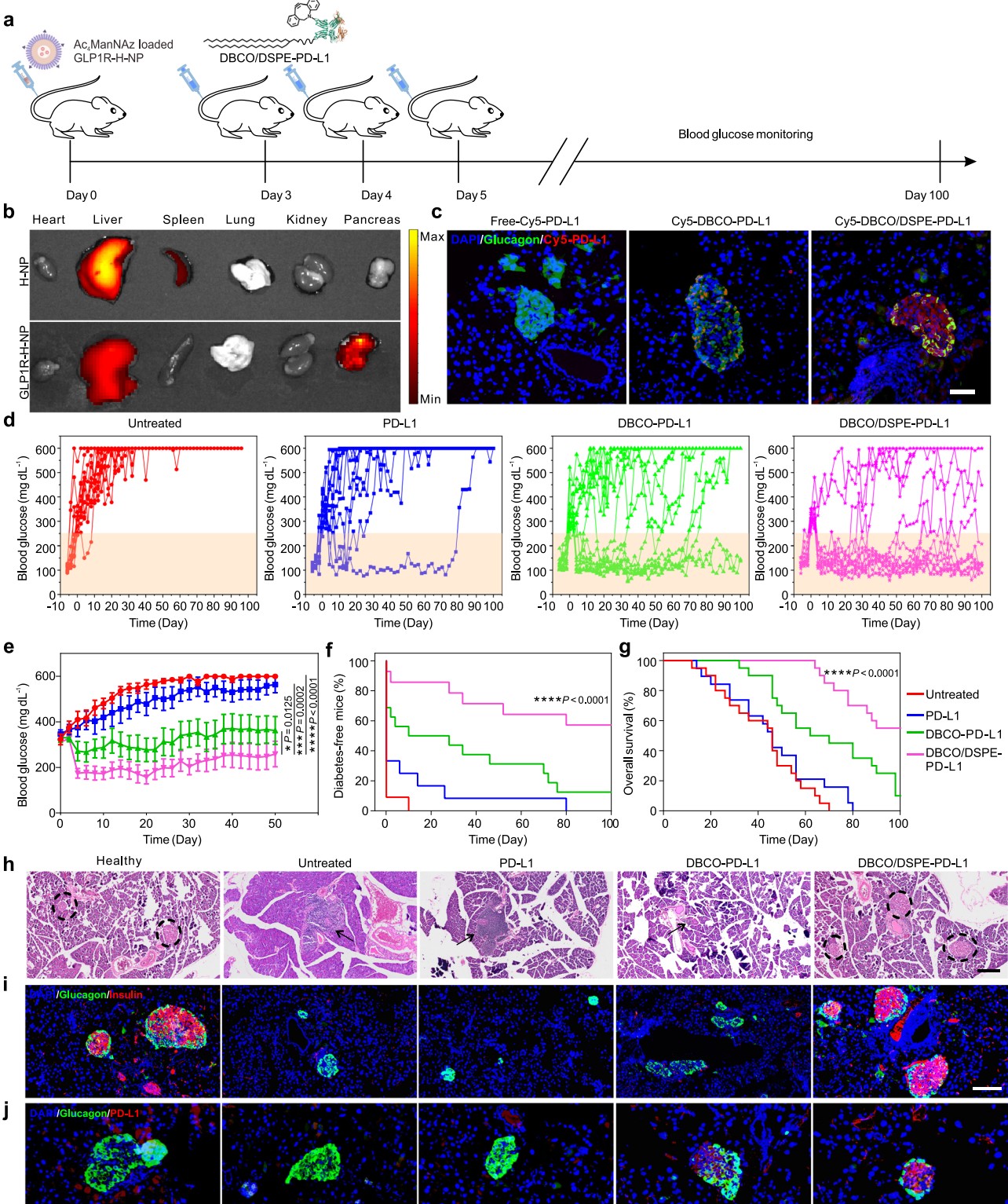

**Fig. 3 | PD-L1 bioengineering reverses the hyperglycemia in the NOD mice.**
**a** Schematic illustration of the treatment process. **b** Ex vivo imaging of main organs revealed that GLP1R-H-NPs could efficiently target the pancreas (6 h post-administration). **c** Fluorescent imaging of the pancreas from NOD mice i.v. injected with Cy5-labeled PD-L1 analogs on day 7. **d** Blood glucose levels of the diabetic NOD mice with different treatments as indicated. **e** Average blood glucose levels of diabetic NOD mice received different treatments. **f** The occurrence of the NOD mice developed diabetes. Data represent the mean ± s.d. (**d**–**f**, $n = 12$ biologically independent samples for untreated group and PD-L1 group, $n = 14$ biologically independent samples

for the DBCO-PDL-1 group and DBCO/DSPE-PD-L1 group.) **g** Survival of the mice received different treatments. Data represent the mean ± s.d. ($n = 20$ biologically independent samples). **h**, **i** Representative H&E stained (**h**) and anti-insulin (red)/anti-glucagon (green) dual stained (**i**) pancreas sections from NOD mice on day 5 post-treatments. Circles indicate islets in the pancreas. Black arrows indicate the infiltration of immune cells in pancreatic islets. Scale bar: 200 μm. **j** Representative anti-PD-L1 (red)/anti-glucagon (green) dual stained pancreas sections from NOD mice on day 5 post-treatments. Scale bar: 50 μm. The data were analyzed by one-way two-sided ANOVA; ****$P < 0.0001$, ***$P < 0.001$, **$P < 0.01$, *$P < 0.05$.

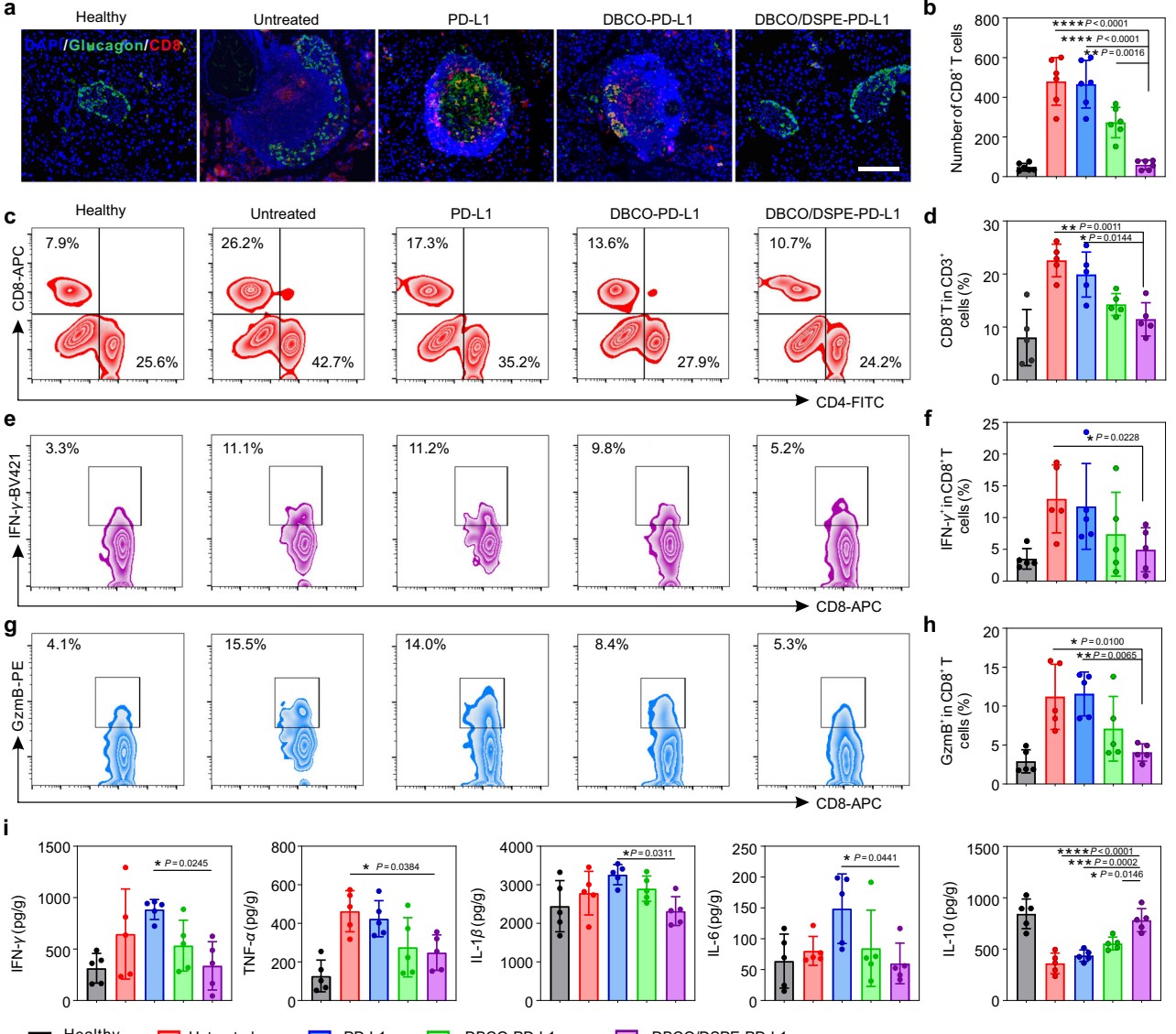

**Fig. 4 | Characterizations of the immunoenvironment in the pancreas of diabetic NOD mice. a, b** Representative anti-glucagon (green)/anti-CD8 (red) dual stained pancreas sections (**a**) and quantification (**b**) from NOD mice on day 5 post-treatments. Scale bar: 200 μm. Data represent the mean ± s.d. ($n$ = 6 biologically independent samples). **c, d** Representative plots of pancreas-infiltrating CD4+ and CD8+ T cells (**c**) and quantification of pancreas-infiltrating CD8+ T cells (**d**) from NOD mice on day 5 post-treatments. Data represent the mean ± s.d. ($n$ = 5 biologically independent samples). **e, f** Representative plots (**e**) and quantifications (**f**) of pancreas-infiltrating CD8+INF-γ+ T cells from NOD mice on day 5 post-treatments by the flow cytometry. Data represent the mean ± s.d. ($n$ = 5 biologically independent samples). **g, h** Representative plots (**g**) and quantifications (**h**) of pancreas-infiltrating CD8+GzmB+ T cells from NOD mice on day 5 post-treatments by the flow cytometry. Data represent the mean ± s.d. ($n$ = 5 biologically independent samples). **i** Cytokines including IFN-γ, TNF-α, IL-1β, IL-6, and IL-10 in pancreas tissues from NOD mice on day 5 post-treatments. Data represent the mean ± s.d. ($n$ = 5 biologically independent samples). The data were analyzed by one-way two-sided ANOVA; ****$P < 0.0001$, ***$P < 0.001$, **$P < 0.01$, *$P < 0.05$.

incubating chondrocytes with Ac4ManNAz, followed by the introduction of PD-L1 analogs. The immobilization of PD-L1 on the cell membrane of chondrocytes was validated through confocal imaging (Supplementary Fig. 30).

We next evaluated the bioengineering efficiency of this dual-anchor coupling approach in a CIA mouse model (Fig. 5a). The Ac4ManNAz-loaded H-NPs were intraarticularly injected in the ankle joints of DBA/1 mice, and then chondrocytes on the surface of articular cartilage could express azido sialic acid derivatives on their membrane. Afterward, Cy5-labeled bovine albumin (BSA) analog as a model protein was administered through tail-vein. As depicted in Fig. 5b, the fluorescence signals of Cy5-BSA analogs appeared 2 h post-administration and reached maximal within 6 h. In addition, over 3.6 times and 1.5 times higher accumulation of Cy5-BSA in the arthrosis of

CIA mice were observed in the Cy5-DBCO/DSPE-BSA-treated group compared to the free Cy5-BSA and Cy5-DBCO-BSA treated groups, respectively (Fig. 5c, d). Notably, the dual-anchoring bioengineering approach extended the restoration of protein on endogenous cells for at least five days.

To explore the therapeutic efficiency of PD-L1 analogs, the signals of joint inflammation in CIA mice were continuously monitored. The mice treated with DBCO/DSPE-PD-L1 exhibited dramatically diminished histopathological inflammation and ankle joint swelling, while the control groups showed continuous deterioration of pathologic features (Fig. 5e). The ankle joints were harvested and evaluated through histomorphometric micro-computed tomography (CT). Severe erosion of ankle joints of arthritis mice was confirmed in the untreated group (Fig. 5f and Supplementary Fig. 31). In comparison,

the ankle joint in the DBCO/DSPE-PD-L1 group showed a smooth surface and normal trabecular structure resembling healthy joints. Paw thickness and arthritis score also showed similar trends (Fig. 5g, Supplementary Figs. 32 and 33).

The progression of RA is closely related to various immunological activities, including the envision of autoreactive T cells and pro-inflammatory macrophages[40,41]. Flow cytometry analysis of immune cell activities and subpopulations revealed a reduction in both CD4[+] and

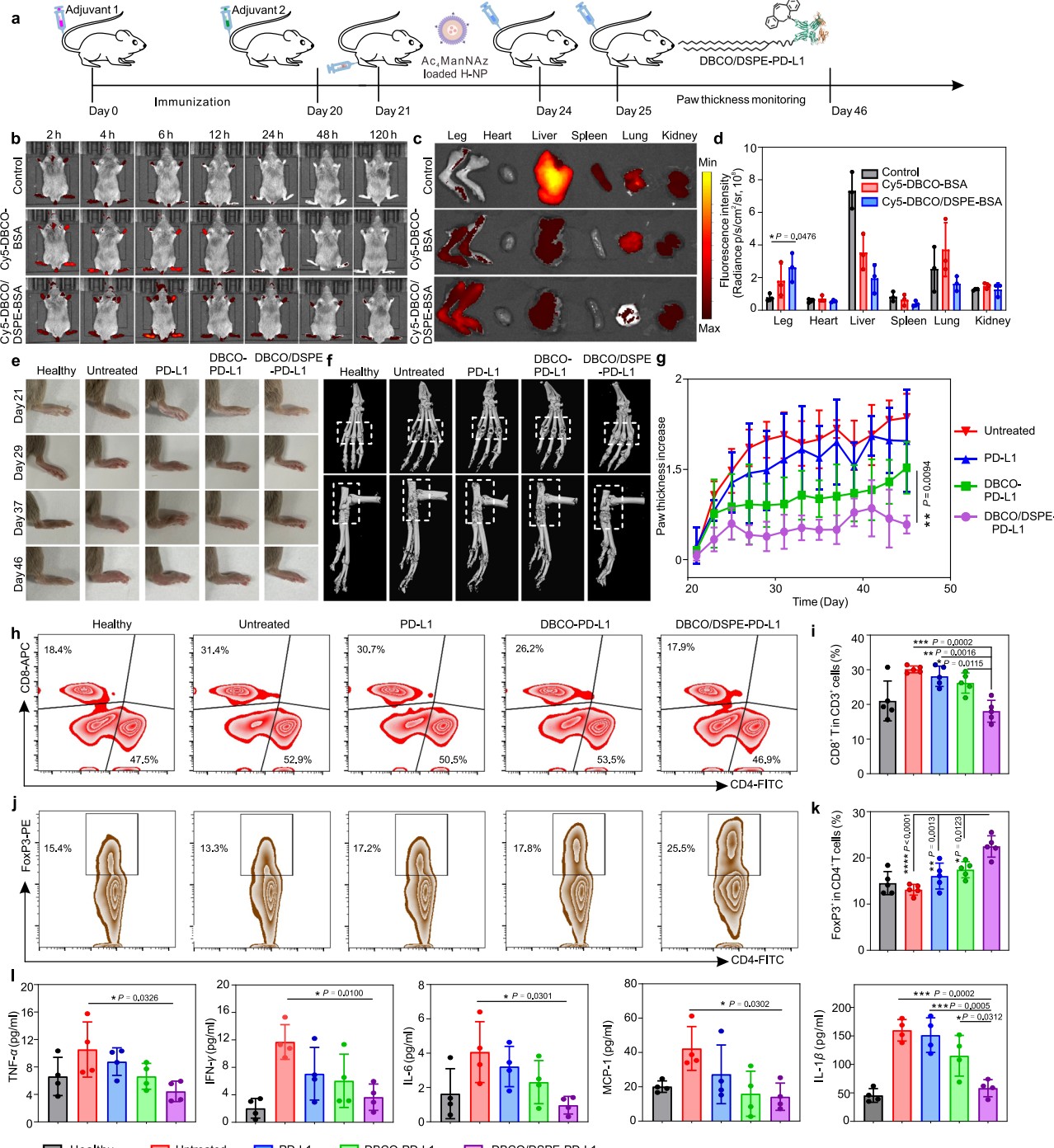

**Fig. 5 | PD-L1 bioengineering for the treatment of CIA. a** Schematic illustration of the treatment process. **b** Representative images of fluorescence distribution in mice with arthritis at different times. **c** Ex vivo fluorescent imaging of main organs and legs. **d** Quantitative analysis of fluorescence intensity in different organs. Data represent the mean ± s.d. (*n* = 3 biologically independent samples). **e** Representative images of hind paw of mice in different treatment groups at the time of each treatment. **f** Representative 3D-reconstructed micro-CT images of right hind ankle joints of normal and arthritic mice after different treatments on day 48. **g** Right hind ankle joint swelling behaviors of arthritic mice. Data represent the mean ± s.d. (*n* = 5 biologically independent samples). **h, i** Representative plots (**h**) and quantifications (**i**) of CD4[+] and CD8[+] T cells in the spleen of mice analyzed by the flow cytometry on day 5 post-treatments. Data represent the mean ± s.d. (*n* = 5 biologically independent samples). **j, k** Representative plots (**j**) and quantifications (**k**) of CD4[+]FoxP3[+] T cells in the spleen of mice from different treatment groups analyzed by the flow cytometry. Data represent the mean ± s.d. (*n* = 5 biologically independent samples). **l** Cytokines including TNF-α, IFN-γ, IL-6, MCP-1 and IL-1β in the serum of different treatment groups analyzed by ELISA. Data represent the mean ± s.d. (*n* = 4 biologically independent samples). The data were analyzed by one-way two-sided ANOVA; \*\*\*\**P* < 0.0001, \*\*\**P* < 0.001, \*\**P* < 0.01, \**P* < 0.05.

CD8[+] T cells upon treatment with PD-L1 analogs after 4 days, indicating the suppression of lymphocytes contributed to symptom alleviation (Fig. 5h, i and Supplementary Fig. 34). The remission was strengthened with prolonged retention of PD-L1 intraarticularly. Moreover, further investigation into their sub-populations revealed decreased cytotoxic T lymphocytes (CD8[+]IFN-$\gamma$[+] and CD8[+]GzmB[+] T cells) and an increased presence of Tregs (CD3[+]CD4[+]FoxP3[+]) in the DBCO/DSPE-PD-L1-treated group (Fig. 5j, k and Supplementary Fig. 35). In addition, a greater number of CD206[+] cells and less CD86[+] cells were observed in the arthrosis of CIA mice treated with PD-L1 analogs, especially DBCO/DSPE-PD-L1, indicating that dual-anchored PD-L1 possessed the ability of inhibiting M1 polarization of synovial macrophages (Supplementary Fig. 36). Furthermore, the production of pro-inflammatory cytokines including TNF-$\alpha$, IFN-$\gamma$, IL-6, MCP−1, and IL-1$\beta$ were reduced (Fig. 5l).

Histopathology analyses were conducted to assess whether modulation of inflammation could inhibit cartilage destruction. H&E staining, safranin-O staining, and Masson's trichrome staining revealed obvious cartilage erosion and bone destruction in the cartilage of CIA mice, with matrix fibrillation extending vertically downward into the mid zone. Such pathology was also observed in mice treated with PD-L1 alone, with only 59.3% of cartilage remaining compared to normal sample (Supplementary Fig. 37). However, the introduction of PD-L1 analogs significantly halted cartilage damage. The mice treated with DBCO/DSPE-PD-L1 showed an intact cartilage surface in both ankles and knuckles that resembled healthy mice (Supplementary Fig. 38). Alleviated synovial hyperplasia and pannus formation were also confirmed in the DBCO/DSPE-PD-L1 group. Finally, the in vivo biosafety of different treatments was assessed by hematological and histological analysis. Insignificant differences of key hematological indicators as well as hepatic and renal parameters were observed (Supplementary Fig. 39). No distinct pathological change was observed after different treatments (Supplementary Fig. 40).

## Discussion

The development of multiple immune-related adverse events in patients receiving anti-PD-L1 therapies emphasizes the significance of PD-L1 in the regulation of autoimmunity[11]. Shifting the PD-1/PD-L1 axis has shown promise in treating autoimmune diseases. In situ immobilization of immune checkpoint on endogenous cells provides a feasible option for enhancing immune tolerance and alleviating autoimmune responses. Current immobilization strategies mainly involve covalent modification, noncovalent attachment, and genetic engineering. Despite significant achievements in improving cell modification efficacy, these approaches have also been restricted by side products of the conjugation reaction, the alteration of cell membrane structure, the inferior specificity towards target cells, or insertional mutagenesis, and the risk of unexpected diseases[42,43].

To this end, we demonstrated an in-situ immobilization strategy for persistent potentiation of PD-L1 on target endogenous cells by taking advantages of both covalent and noncovalent modification approaches. Metabolic glycoengineering followed by bioorthogonal click chemistry enables facile and efficient chemical decoration of biomacromolecules on target cells[22]. In addition, aliphatic chains could be readily inserted into the lipid bilayer of plasma membranes, resulting in steady cell membrane decoration[27]. We substantiated that PD-L1 molecules could be specifically functionalized on the target cell membrane by selective exogenous introduction of an azido group followed by the efficient click reaction. Then the lipid tail of amphiphile conjugates (DBCO/DSPE-PD-L1) was inserted into cell membrane, which significantly prolonged PD-L1 retention over current cell modification methods.

In the early-onset T1D model, this dual-anchor coupling approach exhibited effective immunoregulation by reducing the infiltration of IFN-$\gamma$ and GzmB producing (CD3[+]CD8[+]IFN-$\gamma$[+] and CD3[+]CD8[+]GzmB[+]) cytotoxic T lymphocytes and recruiting more CD3[+]CD4[+]FoxP3[+] Tregs, thereby reversing 57.1% of newly hyperglycemic NOD mice after 100 days. In a proof-of-concept evaluation, we further explored whether such approach could be applied to the treatment of rheumatoid arthritis. Similarly, the dual-anchor coupling approach remitted the deterioration of cartilage in a CIA model by suppressing autoreactive T cells and inhibiting the polarization of synovial macrophages towards pro-inflammatory phenotype. Meanwhile, both bioorthogonal reactions and PEG-DSPE-based peptides have entered clinical trials[28,44], indicating the high translational potential of this protein conjugate. Before the application of this technique to human health, several issues should be thoroughly considered. For example, the length of PEG spacer may influence the efficiency of cell membrane insertion. Although short PEG blocks were reported to show preferential plasma membrane insertion[26], steric hindrance might interfere with cell membrane insertion efficacy of the lipid tails once DBCO/DSPE-PD-L1 reacts with azido sialic acid that locates at the end of glycoprotein. The long-term biosafety evaluation of DBCO/DSPE-PD-L1 for clinical translation should also be thoroughly investigated. In addition, the extension of such an approach to the treatment of other autoimmune diseases that resembles a human treatment situation needs further exploration.

## Methods

### Ethical regulations

All animal handling protocols and experiments were approved by the Guidelines for Care and Use of Laboratory Animals of Zhejiang University (No. ZJU20220109).

### Materials

MPEG (Mn: 5000), 2-bromoisobutyryl bromide, and CuBr and 2, 2'-dipyridine were purchased from Shanghai Aladdin Biochemical Technology Co., LTD (China). Poly(vinyl alcohol) (PVA, 89-98 KDa), 2-(dimethylamino)ethyl methacrylate (DMAEMA), 4-(bromomethyl) phenylboronic acid were provided by Sigma-Aldrich Co. (USA). $Ac_4ManNAz$ and DBCO-NHS were obtained from MedChemExpress. Tetrahydrofuran (THF), triethylamine (TEA), $N, N$-Dimethylformamide (DMF) were obtained from Shanghai Macklin Biochemical Co., Ltd (China). PBS (pH 7.4), fetal bovine serum (FBS), dulbecco's modified eagle medium (DMEM), trypsin, ethylene diamine tetraacetic acid (EDTA), and penicillin-streptomycin were purchased from Gibco Co., Ltd. (USA). All of the antibodies (Anti-CD3, CD4, CD8, Foxp3, GrzmB, and IFN-$\gamma$) used for fluorescence-activated cell sorting (FACS) were purchased from Biolegend Inc. All chemical agents were of analytical grade and were used directly without further purification.

### Synthesis of MPEG$_{5k}$-P(DMAEMA)$_{6k}$

MPEG-Br was first prepared. MPEG (Mn: 5000, 10.0 g) was dissolved in anhydrous THF (30 mL) in the ice-water bath. Then TEA (10 mmol, 1.39 mL) and 2-bromoisobutyryl bromide (10 mmol, 0.62 mL) were introduced into the MPEG solution with magnetic stirring. After a 2-h reaction at 0 °C and further stirring under ambient temperature for about 22 h, the insoluble salts were removed by filtration and the THF solvent was concentrated in an evaporator. The resulting polymer was achieved using diethyl ether and dried under 40 °C in a vacuum for 24 h to obtain a white powder of MPEG-Br.

MPEG$_{5k}$-P(DMAEMA)$_{6k}$ was synthesized *via* the atom transfer radical polymerization (ATRP) method[21]. In detail, MPEG$_{5k}$-Br (0.4 g), CuBr (11.4 mg), and 2, 2'-dipyridine (25 mg) were introduced into a 20 mL round bottom flask under $N_2$ protection. Then, anhydrous THF (4 mL) and DMAEMA (0.4 g) were added to the mixture and gently mixed. The flask was sealed with $N_2$ and immersed in an oil bath after three freeze-thaw cycles. The reaction continued to be stirred at 60 °C overnight. The resultant solution was further purified by ethyl acetate (200 mL) and $NaHCO_3$ (1 M, 3 × 100 mL) and dried with anhydrous $Na_2SO_4$. Finally, slightly yellow viscous solid products were obtained after filtration and desiccation.

## Synthesis of MPEG$_{5k}$-P(DMAEMA-PBA)$_{6k}$

The synthesized MPEG$_{5k}$-P(DMAEMA)$_{6k}$ (0.22 g) was dissolved in DMF, and 0.5 g of 4-(bromomethyl) phenylboronic acid was added into the solution. Then, the mixture was continuously stirred at 60 °C overnight and purified by dialysis. The white product was obtained after filtration and lyophilization.

## Preparation of Ac$_4$ManNAz loaded ROS-responsive nanoparticle

In a typical reaction, Ac$_4$ManNAz (5 mg), MPEG$_{5k}$-P(DMAEMA)$_{6k}$ (1 mg mL$^{-1}$, 2.5 mL), and NHS-PEG$_{5k}$-PLGA$_{6k}$ (1 mg mL$^{-1}$, 2.5 mL) were mixed. Ac$_4$ManNAz-loaded ROS-responsive nanoparticles were prepared via the nanoprecipitation method. In order to stabilize the formed nanoparticles, PVA (1 wt%, 0.5 mL) was introduced.

## Preparation of DBCO and DBCO/DSPE conjugated PD-L1

DBCO conjugated PD-L1 was prepared through a typical amine-NHS coupling reaction. PD-L1 was dispersed in PBS solution (3.8 nM, 200 μL, 1 mg mL$^{-1}$) and the pH was adjusted to 8.0 by 0.1 M of sodium bicarbonate solution. Then, DBCO-NHS (150 nM, 3 μL, 20 mg mL$^{-1}$ in DMSO) was introduced to the solution. The reaction continued for 2 h under gentle shaking at room temperature. The DBCO conjugated PD-L1 (DBCO-PD-L1) was further purified three times by ultrafiltration (10 K MWCO, 4000 rpm, 20 min).

DBCO/DSPE conjugated PD-L1 was fabricated by introducing maleimide (Mal) functionalized DSPE into the prepared DBCO-PD-L1. Similarly, DBCO-PD-L1 was dispersed in PBS solution (3.7 nM, 200 μL, 1 mg mL$^{-1}$) and reacted with DSPE-PEG$_{2k}$-Mal (74 nM, 210 μL, 1 mg mL$^{-1}$ in DMSO). The reaction continued for 4 h under gentle shaking at room temperature. Then, DBCO/DSPE conjugated PD-L1 (DBCO/DSPE-PD-L1) was further purified by ultrafiltration (50 K MWCO, 4000 rpm, 20 min).

## Cell culture and in vitro cytotoxicity analysis

Min 6 cells were maintained in RPMI 1640 medium containing 10% FBS, 100 IU mL$^{-1}$ penicillin, and 100 μg mL$^{-1}$ streptomycin. Chondrocytes were incubated in DMEM (high glucose) medium with 10% FBS, 100 IU mL$^{-1}$ penicillin, and 100 μg mL$^{-1}$ streptomycin. All the cells were maintained at 37 °C in a 95% air, 5% CO$_2$ atmosphere.

The Min 6 cells were seeded at a density of $1 \times 10^4$ cells per well in a 96-well plate. The cytotoxicity of Ac$_4$ManNAz and H-NPs with different concentrations was quantified by CCK-8 assay according to the manufacturer's protocol.

## Preparation of azido-expressed Min 6 cells

Azido-expressed Min 6 cells were prepared by incubating cells ($6 \times 10^6$ cells per 10 cm dish) with 50 μM of Ac$_4$ManNAz in a complete growth medium for 3 days. The Ac$_4$ManNAz-containing medium was refreshed every 24 h. Then, azido-expressed Min 6 cells were detached by enzyme-free cell dissociation buffer according to the manufacturer's protocol.

## Immobilization of PD-L1 analogs onto azido-expressed Min 6 cells

To visualize the immobilization of PD-L1 analogs on Min 6 cells, PD-L1 molecules were first labeled with fluorescein isothiocyanate isomer (FITC) through a typical amine-NHS coupling reaction[13]. Then FITC-labeled PD-L1 (FITC-PD-L1) was used for further functionalization to obtain PD-L1 analogs (FITC-DBCO-PDL1 and FITC-DBCO/DSPE-PD-L1). The immobilization was carried out by introducing FITC-DBCO-PDL1 and FITC-DBCO/DSPE-PD-L1 (10 nM) into azido-expressed Min 6 cells ($1 \times 10^5$ cells). The bioorthogonal reaction and physical insertion were allowed to progress at 37 °C in the serum-free RPMI-1640 medium for 1 h. The PD-L1 immobilized cells were washed twice with serum-free RPMI-1640 medium before further studies.

## In vitro T cell binding and activity assay

Lymph node T cells were isolated from NOD mice using a T cell isolation kit. Then, the cells were grown in RPMI 1640 medium containing 2% FBS and 2 μg mL$^{-1}$ anti-CD3 and 5 μg mL$^{-1}$ anti-CD28 in 5% CO$_2$ at 37 °C for 48 h. PD-L1 functionalized Min 6 cells and free Min 6 cells ($2 \times 10^4$) were incubated with CD3$^+$ T cells ($1 \times 10^6$) for 24 h, respectively. Then, the cells were fixed with 4% paraformaldehyde for 10 min. Min 6 cells and T cells were stained with anti-PD-L1 and anti-PD-1 antibodies, respectively. The nucleus was stained with DAPI for 10 min. The binding of T cells with Min 6 pancreatic cells was observed by a confocal microscope. To further investigate the activity of T cells, the T cells were collected and analyzed through flow cytometry.

## Treg cell suppression assay

The in vitro suppression assay of CD4$^+$FoxP3$^+$ cells was conducted as follow: Lymph node CD3$^+$ T cells were isolated from NOD mice using a T cell isolation kit. PD-L1 functionalized Min 6 cells were ($2 \times 10^4$) incubated with CD3$^+$ T cells ($1 \times 10^6$) for 24 h. Then, Treg cells were isolated and purified from the spleen collected CD3$^+$ T cells by EasySep Mouse CD4$^+$CD25$^+$ Regulatory T Cell Isolation Kit II and Mouse T Cell Isolation Kit (STEMCELL Technologies) as the manufacturer instructed. All cells were cultured in the complete T cells culture medium (RPMI 1640 with 10% heat-inactivated FBS, 1× penicillin/streptomycin, 1× sodium pyruvate, 1× nonessential amino acids, 20 mM HEPES, and 50 μM β-mercaptoethanol) with IL-2 (30 IU/ml; PeproTech, catalog no. 200-02) for CD3$^+$ conventional T (Tcon) cells and IL-2 (1500 IU/ml) for Tregs[45]. The CellTrace™ Cell Proliferation Kits (Introgen, catalog no. C34571) were applied for monitoring cell proliferation of Tcon cells. For the Treg suppression assay, $10^5$ CellTrace™ stained Tcon cells were incubated with different amount of Treg cells ($10^5$, $5 \times 10^4$, and $2.5 \times 10^4$, respectively) in an anti-CD3/CD28-coated 48-well plate for 5 days. Then, the cells were collected for flow cytometry. The suppressive capacity of Tregs was calculated as proliferation index (the number of proliferated cells to the unproliferated cells).

## In vivo biodistribution of H-NPs in NOD mice

Cy7.5 labeled ROS-responsive H-NPs and GLP1R-H-NPs were injected into NOD mice through tail-vein, respectively. The mice were euthanized at different time points, respectively. The main organs including the pancreas, heart, liver, spleen, lung, and kidney were collected. Caliper IVIS spectrum imaging system was utilized to monitor and quantify the fluorescence intensity of the organs.

## Diabetic NOD mice treatment

NOD/ShiLtJ female mice (8 weeks old) were purchased from Huafukang. Blood glucose was monitored every day until the blood glucose levels were above 250 mg dL$^{-1}$ for two consecutive days. Then, the hyperglycemia mice were left untreated as a control group or injected with free PD-L1 (PD-L1 group) via tail-vein three times. In addition, other hyperglycemia mice were injected with Ac$_4$ManNAz-loaded ROS-responsive GLP1R-H-NPs (200 μg of Ac$_4$ManNAz per mouse) via tail-vein. Then, DBCO-PD-L1 or DBCO/DSPE-PD-L1 (50 μg of PD-L1 per mouse) was administered three days after the injection of Ac$_4$ManNAz. This process was repeated three times. The blood glucose level was monitored every two days to the endpoint.

## In vivo T cells analysis

In order to evaluate the status of the infiltrating T cells, the pancreas, lymph node, and spleen were collected from NOD mice with different treatments (3 day and 10-day post-treatments). The cells were dissociated to generate single-cell. Then the samples were collected after passing through a 70-μm filter. Subsequently, the cells were analyzed via flow cytometry.

## Dual-anchor coupling strategy prolonged protein retention in the ankle joint of DBA/1 mice

The $Ac_4ManNAz$-loaded H-NPs were intraarticularly injected in the ankle joints of DBA/1 mice. Then, Cy5-labeled model protein bovine albumin (BSA) analogs (Cy5-BSA, Cy5-DBCO-BSA, and Cy5-DBCO/DSPE-BSA) were injected through tail-vein, respectively. The mice were euthanized at different time points (respectively), and the main organs including the pancreas, heart, liver, spleen, lung, kidney and back ankle joints were collected. Caliper IVIS spectrum imaging system was utilized to monitor and quantify the fluorescence intensity of the organs.

## RA treatment

The autoimmune arthritis model was induced by immunization with an emulsion of Freund's adjuvant and type II collagen (CII)[46]. Specifically, male DBA/1 mice (8 weeks old) were injected with chicken type II collagen emulsified in complete Freund/s adjuvant (CFA), followed by boosting 20 days later with chicken type II collagen emulsified in incomplete Freund/s adjuvant (IFA). The mice were divided into four groups: (1) untreated control group; (2) free PD-L1 administration (PD-L1) group; (3–4) joint injection of $Ac_4ManNAz$-loaded ROS-responsive H-NPs (200 μg of $Ac_4ManNAz$ per mouse), followed by the administration of DBCO-PD-L1 (3) or DBCO/DSPE-PD-L1 (50 μg of PD-L1 per mouse) (4) three days after the injection of $Ac_4ManNAz$. The paw thickness and the arthritis score were evaluated every two days to record the development of arthritis. The level of inflammation for each paw was graded from 0 to 4 by the following scale: absence of inflammation = 0, paw with detectable swelling in a single digit = 1, paw with swelling in more than one digit = 2, paw with swelling of all digits and instep = 3. Severe swelling of the paw and ankle = 4.

## Micro-computed tomography (CT)

On day 46, the DBA/1 mice were sacrificed and their back ankle joint tissues were harvested. Then, micro-CT (Bruker, SKYSCAN1272) was applied to reconstruct the 3D structure of ankle joints and evaluate the histomorphometric characteristics.

## Flow cytometry

Cultured cells were harvested or separated from tissues to obtain a single-cell suspension and treated with red blood cell lysis buffer, then washed with cold PBS two times. Thereafter, the cells were stained with a combination of fluorescence-conjugated mouse antibodies. For cell surface staining, cells were blocked by CD16/32 for 10 min and then incubated with antibodies of PerCP/Cyanine5.5-CD3, FITC-CD4, APC-CD8a, PE/Cyanine CD49b, and Brilliant Violent421 CD25 for 30 min in dark at room temperature. For intracellular staining, cells were pre-treated with a cell activation cocktail (with Brefeldin A) for 4–6 h. Then, cells were fixed in the Fixation Buffer for 30 min at room temperature. After being washed with Perm/Wash buffer, cells were incubated with antibodies of Brilliant Violent421-IFN-γ and PE-Granzyme B for 30 min at room temperature. For intra-nuclear staining, cells were fixed with True-Nuclear™ 1X Fix Concentrate for 60 min in dark at room temperature. After being washed with True-Nuclear™ 1X Perm Buffer, cells were stained with PE-FoxP3 for 30 min at room temperature. Then all samples were washed three times with cold PBS and analyzed by Beckman CytoFlex (CytoFlex S) flow cytometer and FlowJo.

## Histological analysis

The pancreas was collected and fixed in 4% formaldehyde, then embedded in paraffin. The ankle joints were decalcified after fixation. The samples were cross-sectioned into 5 μm thick slices. Tissue slices were blocked and permeabilized for 1 h in the blocking buffer (0.2% Triton-X100 in 3% BSA). Thereafter, the slices were incubated with primary antibodies (CD3, CD4, CD8, FoxP3, Glucagon, Insulin, CD206, and CD86) overnight at 4 °C. After washing with PBS three times, the slices were incubated in the blocking buffer containing corresponding fluorophore-conjugated secondary antibodies for 1 h at room temperature. The nucleus was stained with DAPI for 10 min. Slices were then mounted with mounting media right away, and images were taken within a week[47].

## Statistical and reproducibility

All the experimental data were statistically analyzed and the results were expressed as a mean ± standard deviation (s.d.), $n \geq 3$. GraphPad (version number: 9.2.0) software was used for statistical analysis. Statistical significance was calculated *via* a two-tailed Student's *t*-test for two-group comparisons. Statistical differences were determined using one-way two-sided analysis of variance (ANOVA) for multiple comparisons. Statistical significance was set as follows: $*P < 0.05$, $**P < 0.01$, $***P < 0.001$, $****P < 0.0001$, and ns denotes no significant difference. For the data in Figs. 1b, e; 2f; 3b, c, h–j; 4a, 5b, c, e, f and Supplementary Figs. 4c; 7a; 9a, c; 13a; 15a, b; 16a; 20; 23; 30; 31; 32a; 34; 36; 37; 38; and 40, three experiments were repeated independently with similar results, and results from representative experiments were shown.

## Reporting summary

Further information on research design is available in the Nature Portfolio Reporting Summary linked to this article.

## Data availability

The authors declare that all the data supporting the findings of this study are available within the article and supplementary information and from the corresponding authors upon request. Source data are provided as a Source Data file. Source data are provided with this paper.

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

## Acknowledgements

This work was supported by the grants from National Natural Science Foundation of China (32271380) to J.Y., the National Key R&D Program of China (2021YFA0909900) and Zhejiang Province "Kunpeng Action" Plan to Z.G., the National Natural Science Foundation of China (32101064), Fundamental Research Funds for the Central Universities (2021FZZX001-47) to Y.Z., the Startup Packages of Zhejiang University to Z.G., J.Y., and Y.Z., and China Postdoctoral Science Foundation (2021TQ0280) to S.W.

## Author contributions

J.Y., Z.G., and S.W. conceived the project. S.W., Ying Z., Y.W., and S.Z. performed the experiments and collected the data. Y.Y. assisted in the animal experiments. T.S. drew the scheme and polished the figures. S.W. analyzed the data. S.W., Y.Z., Z.G., J.W., and J.Y. prepared the manuscript. All the authors discussed the results and commented on the manuscript.

## Competing interests

Z.G., J.Y., and S.W. have applied for a patent related to this work. Z.G. is the co-founder of Zenomics Inc. and ZCapsule Inc. Z.G. and Y.Z. are the co-founders of μZen Pharma Co., Ltd., and the other authors declare no competing interests.
