## [Peer Review File · Nature Communications]

An In Situ Dual-Anchoring Strategy for Enhanced Immobilization of PD-L1 to Treat Autoimmune DiseasesREVIEWER COMMENTS

Reviewer #1 (Remarks to the Author):

The manuscript by Wang and colleagues reports a two-step dual anchor coupling strategy to immobilize PDL-1, a checkpoint molecule key to immunotolerance, on endogenous target cells. This is achieved using a combination of biorthogonal chemistry and physical insertion of the cell membrane. The proposed strategy is applied to experimental mouse models of type 1 diabetes (T1D) and rheumatoid arthritis, resulting in decreased autoreactive lymphocytes, increased FOXP3⁺ cells and ameliorated disease clinical course. Disease improvement is reflected by maintained glucose homeostasis for over 100 days in T1D and reduced cartilage erosion and bone destruction in the collagen-induced arthritis model. The proposed approach is sound with methodological details clearly described. The experiments are well developed and outlined in a logical way. There are, however, concerns about the strength of some of the data presented that would require consideration and further validation.

Detailed comments are as follows:

- Fig 2D: When considering fluorescence on Day 3, no substantial difference is noted when comparing DBCO/DSPE-PDL-1 and DBCO-PDL-1 labeled Min-6 cells. Since differences at this time point are evident in Fig. 2C, the Authors should consider including a more representative staining.
- Fig. 2G: The Authors show flow cytometry data indicating a decrease in IFN γ ⁺ and granzyme B⁺ CD8 cells and an increase in CD4⁺FoxP3⁺ lymphocytes in DBCO-DSPE/PDL-1 labeled cells, obtained from the lymph nodes of hyperglycemic NOD mice. Were the functional properties of these cells tested? Were CD4⁺Foxp3⁺ cells tested in an in vitro suppression assay?
- Is the stabilization of PDL-1 on CD8 and Treg cells favoring immune exhaustion when a prolonged binding is achieved?
- Fig. 3B: there is a high fluorescence in the liver of mice injected with both H-NP and GLP1R-H-NP. Was the liver of these mice examined for histology, T cell infiltration and biochemical markers of inflammation (i.e., AST, ALT)?
- Fig. 3D: A large variability in blood glucose levels is noted in mice undergoing DBCO/DSPE-PDL-1 treatment.
- Fig. 3H: Increase in the number of islets cannot be clearly appreciated in the DBCO/DSPE-PDL-1 treated mice. The Authors should consider including a more representative, convincing picture.
- Fig. 4: Changes in CD8 T cell phenotype (i.e., significantly lower IFN γ , granzyme B and cytokine levels) are noted when comparing G3 or G2 with G5. However, no significant differences are noted between G4 and G5.
- Supplementary Fig. 16: Have the Authors measured cytokine producing cells among CD4 lymphocytes? Is the beneficial effect in the DBCO/DSPE-PDL-1 group linked to increased Tregs or decreased CD8 effectors?
- Are systemic (i.e., peripheral blood/spleen) and local (i.e., pancreatic lymph nodes) compartments impacted in DBCO-PDL-1 and DBCO/DSPE-PDL-1 groups?
- Supplementary Fig. 17: Foxp3 staining is almost unnoticeable. The Authors should provide a more representative picture.
- Fig. 5D: The cumulative data presented do not adequately reflect the representative image, when considering lung, kidney, and spleen.
- Fig. 5G: The Authors should check the data represented in this graph because G5 appears to have the highest paw thickness increase. The same applies to Supplementary Fig. 22B.
- As in the T1D model, are the beneficial effects noted in G5 linked to decreased effectors or

increased Tregs?

- Supplementary Fig. 29-30: There are rather high AST and ALT levels despite normal appearing liver histology. How do the Authors explain these findings?
- Discussion: The Authors could elaborate on the advantages of the proposed approach over immunotherapies that are currently used or are in development. Limitations of the proposed approach should be mentioned.

Minor points

- Line 89: 'indistinctive' should be replaced with 'negligible'.
- Line 252: please replace 'insignificant' with 'non-significant'.

Reviewer #2 (Remarks to the Author):

The manuscript of Wang et al. presents an approach aimed at increasing the density of expression of a checkpoint inhibitor ligand namely, the PD-L1 molecule within sites that are the target of an autoimmune reaction. In autoimmune diabetes, one of models analyzed the strategy should help counteracting the pathogenic capacity of CD4+ and CD8+ autoreactive T lymphocytes. The technical approach to increase the expression of the target molecule in a given cell is based on metabolic glycoengineering and bioorthogonal click chemistry to "anchor" the selected checkpoint inhibitor.

MAJOR COMMENTS

1) The strategy presented is not innovative per se; what is novel is the cell type the authors chose to target using the glycoengineered PD-L1 that is the insulin-producing beta-cell within the pancreas of the host. Not being a biochemist myself, I cannot judge on the method used to ensure that the coupling of the chosen molecule (that is in fact a modified form of PD-L1 in order to extend its half-life on beta-cells of the islets of Langerhans) was successful. I will therefore limit myself to reviewing the in vitro and in vivo biological data reported by the authors and their discussion on the importance of their findings in the context of the state of the art.

To that aim, it is important to consider and confront the model and the results presented in this manuscript to the recent results published by the group of R. Tisch in *Advanced Materials*, 2021 (this paper is in fact referenced in the present manuscript as #13 (Au et al.)). This other group described the bioengineering (glycoengineering and bioorthogonal click chemistry) of a beta-cell line NIT-1 (established from NOD islets) to achieve increased surface expression of a combination of 3 checkpoint inhibitor ligands: PDL1, CD86 and Galectin69. The modified cell lines were placed into an acellular scaffold which was implanted subcutaneously in overtly diabetic NOD mice. The device could reproduce "a pancreas microenvironment" where diabetogenic T lymphocytes migrated and were, according to the authors "neutralized" via exhaustion, thereby reversing recent onset type 1 diabetes for about 2 months.

Here the authors take a step further by proposing to couple the modified PD-L1 molecule directly to the beta-cells of the host using as a specific anchor molecule a ligand of the glucagon-like peptide-1 receptor (GLP1R). The therapeutic compound may therefore be administered quite simply by the i.v. route. All this to underscore that the innovation in this manuscript concerns the intrinsic characteristics of the therapeutic molecule: -which contains only PD-L1 (instead of the 3 molecules PD-L1, CD86 and Galectin 9 in the paper by Au et al) and -which includes a molecule allowing the direct targeting of beta-cells of the host, hence its simple administration. It is therefore necessary to closely analyze whether the efficacy

criteria of the therapeutic compound are really significant. In vivo data upon administration in diabetic NOD mice (Figures 3d and e) would argue for a significant effect quite similar to that in the manuscript of Au et al. The major problem is however that the cellular, histological data and, above all, confocal microscopy data do not show sufficiently clear images to conclude: - neither on the precise localization of the therapeutic product in the pancreas, - nor on the images characterizing the lymphocyte infiltrate in the pancreas, -nor on images of insulinitis in conventional histology. In particular, Fig. 3c, Fig 3i, Supplementary Fig 11a, Supplementary Fig 15, Supplementary Fig 17 should be revisited. Also, in Fig 2f the “colocalization of Min 6 cells or PD-L1 conjugated Min 6 cells and lymph node T cells were displayed by immunofluorescence staining” is really not easy to see; PD-1 red spots do not look like lymphocytes with a membrane, a cytoplasm, a nucleus. Higher magnifications and clear superimposed images are needed.

2) I am puzzled by control data in figure 3g: how could the authors maintain untreated diabetic NOD mice for a median time of about 60 days. In colonies with a reasonable diabetes incidence >60% in females once diabetic animals die or are culled within 2 weeks since they lose weight and waste away very quickly. What is the incidence of diabetes in the NOD females used?

3) Data in collagen-induced arthritis appear preliminary. The authors should concentrate on autoimmune diabetes.

Reviewer #3 (Remarks to the Author):

In this manuscript by Wang et al., the authors describe a novel technique for anchoring exogenous PD-L1 on the membrane of target cells within in mouse, treating autoimmunity in two disease models. These are intriguing data with convincing in vivo results. However, the correlative data do not sufficiently convince me that this technique is working through the mechanism that the author’s propose. I have the following concerns.

General concerns regarding impact

1. The authors show that their technique relatively specifically localizes PD-L1 to the pancreas. Their data are not convincing that they can do the same for the joints. The authors have demonstrated some proof of principle in one model, but without demonstrating more specificity for their labeling process, I am not convinced that this technique will necessarily be an improvement over broad immunosuppression. This does not necessarily negate the value of their findings, but it does warrant discussion in the text as a limitation.
2. From the data as presented, it is not clear how long the PD-L1 conjugate will remain in place and how long the immune suppressing effects will last. The impact would be greatly enhanced were much later time points (4 months, 6 months) examined, particularly for persistence of the exogenous PD-L1. Again, this does not necessarily mean that the current manuscript is insufficient, but it is a limitation of the data as provided.

Concerns with the Results

3. Although using FITC conjugated DBCO/DSPE-PD-L1 demonstrates that the construct is associated with the target cells of interest, it is important to determine whether this PD-L1 is appropriately folded. Using fluorophore conjugated PD-1 as a probe as well as a fluorophore conjugated anti-PD-L1 antibody would both be valuable to assess surface expression of folded PD-L1 capable of associating with PD-1.
4. Figure 1d looks like DBCO/DSPE-PD-L1 loses fluorescence faster than DBCO-PD-L1,

perhaps I am missing something, but this appears to be the opposite of the author's claim. I suspect that this is a typo.

5. The "G" labeling on this figure is confusing. I recommend having a clear legend with the treatment groups visible near the bar graphs. Color coding is sufficient to identify the groups if a legend is present.

6. I would like anti-PD-1 treated cultures or PD-1 KO T cells as an additional control for figure 2 g and h. It would also be valuable to have a negative control cell line as a target that you would not expect these T cells to activate in the presence of. This cell line with or without DBCO/DSPE-PD-L1 could be exposed to T cells; additional controls would not be necessary provided that these two conditions were negative.

7. The data in 3h and 3i need to be quantified in the main figure. This would also be much more convincing if it contained PD-L1 immunofluorescence. PD-L1 is typically expressed in the NOD pancreas, so it would be important to demonstrate that the total PD-L1 level is actually increased by this treatment in order to confirm the proposed mechanism of action.

8. The BSA conjugation seems much less specific overall than the GLP1R conjugate. Although the authors do not see toxicity from this, this is almost certainly because these mice were not undergoing an infectious challenge (something that will certainly not be the case with free living people). The authors should acknowledge this lack of specificity and provide more of a rationale for BSA conjugation than they currently do.

9. I think that figure 4g is mislabeled using the "G" labeling format. If not, then these data appear to show that the DBCO/DSPE-PD-L1 construct actually makes the disease worse?

General Minor Comments

10. In general, the English needs some work throughout the manuscript. Several of the sentences have an odd structure that is difficult to follow.

11. The title is not accurate. The authors only used PD-L1, thus "Immune Checkpoints" is more general than they have shown.

12. "and inflammatory bowel disease developed from the imbalance of the immune systems," this is an over simplification. We really don't know why these diseases develop.

13. "the various side effects and long-term medication have severely reduced the quality of life for patients" the quality of life impact of many immunosuppressants is relatively small. The main drawback is often that the medications are not sufficiently effective, particularly for a disease like Type 1 Diabetes, where they have no role.

Responses to reviewer's comments

We sincerely thank the reviewers for their valuable comments and suggestions. Below we have provided responses to the comments and have accordingly revised the manuscript.

Reviewer #1:

The manuscript by Wang and colleagues reports a two-step dual anchor coupling strategy to immobilize PDL-1, a checkpoint molecule key to immunotolerance, on endogenous target cells. This is achieved using a combination of biorthogonal chemistry and physical insertion of the cell membrane. The proposed strategy is applied to experimental mouse models of type 1 diabetes (T1D) and rheumatoid arthritis, resulting in decreased autoreactive lymphocytes, increased FOXP3⁺ cells and ameliorated disease clinical course. Disease improvement is reflected by maintained glucose homeostasis for over 100 days in T1D and reduced cartilage erosion and bone destruction in the collagen-induced arthritis model.

The proposed approach is sound with methodological details clearly described. The experiments are well developed and outlined in a logical way. There are, however, concerns about the strength of some of the data presented that would require consideration and further validation.

Detailed comments are as follows:

1. Fig 2D: When considering fluorescence on Day 3, no substantial difference is noted when comparing DBCO/DSPE-PDL-1 and DBCO-PDL-1 labeled Min-6 cells. Since differences at this time point are evident in Fig. 2C, the Authors should consider including a more representative staining.

Response:

Thanks so much for the reviewer's valuable suggestion. In Fig. 2b, the fluorescent signal of PD-L1 (green color) on DBCO/DSPE-PDL-1 treated Min 6 cells was much stronger than the DBCO-PDL-1 treated cells, which was consistent with the quantitative result in Fig. 2c. In addition, we have also conducted a quantitative analysis of the flow cytometry assay, and found that the ratio of PD-L1⁺ Min 6 cells in the DBCO/DSPE-PDL-1 group was 81.9% on Day 3, which was higher than that of the DBCO-PDL-1 group (56.1%) (Fig. L1). We have added the quantitative information in Fig. 2d and Fig. S7c in the revision.

Fig. L1. The dual-anchor coupling strategy prolonged the immobilization of PD-L1 on the cell membrane. **a** Flow cytometry of Min 6 cells labelled with FITC-PD-L1 analogs at predetermined time points. **b** Quantitative assay of Min 6 cells labeled with PD-L1 analogs on Day 3 *via* flow cytometry. Data represent the mean \pm s.d. (n = 3).

2. Fig. 2G: The Authors show flow cytometry data indicating a decrease in IFN γ ⁺ and granzyme B⁺ CD8 cells and an increase in CD4⁺FoxP3⁺ lymphocytes in DBCO-DSPE/PDL-1 labeled cells, obtained from the lymph nodes of hyperglycemic NOD mice. Were the functional properties of these cells tested? Were CD4⁺Foxp3⁺ cells tested in an *in vitro* suppression assay?

Response:

The *in vitro* suppression assay of CD4⁺FoxP3⁺ cells was conducted as follow: Lymph node CD3⁺ T cells were isolated from NOD mice using a T cell isolation kit. PD-L1 functionalized Min 6 cells were (2×10^4) incubated with CD3⁺ T cells (1×10^6) for 24 h. Then, regulatory T (Treg) cells were purified from the collected CD3⁺ T cells by EasySep Mouse CD4⁺CD25⁺ Regulatory T Cell Isolation Kit II and Mouse T Cell Isolation Kit (STEMCELL Technologies) as the manufacturer instructed. All cells were cultured in the complete T cells culture medium (RPMI 1640 with 10% heat-inactivated FBS, 1 \times penicillin/streptomycin, 1 \times sodium pyruvate, 1 \times nonessential amino acids, 20 mM HEPES, and 50 μ M β -mercaptoethanol) with IL-2 (30 IU/ml; PeproTech, catalog no. 200-02) for CD3⁺ conventional T (Tcon) cells and IL-2 (1500 IU/ml) for Treg cells¹. The CellTrace™ Cell Proliferation Kits (Introgen, catalog no. C34571) were applied for monitoring cell proliferation of Tcon cells. For the Treg suppression assay, 10^5 CellTrace™ stained Tcon cells were incubated with different amount of Treg cells (10^5 , 5×10^4 , and 2.5×10^4 , respectively) in an anti-CD3/CD28-coated 48-well plate for 5 days.

Then, the cells were collected for flow cytometry. The suppressive capacity of Tregs was calculated as proliferation index (the number of proliferated cells to the unproliferated cells). As shown in Fig. L2, the suppressive functionality of Tregs was confirmed through the *in vitro* suppression assay.

Reference:

1. Zhang, W. *et al.* Adoptive Treg therapy with metabolic intervention via perforated microneedles ameliorates psoriasis syndrome. *Sci. Adv.* **9**, eadg6007 (2023).

Fig. L2. Treg cell suppressive activity. a,b, Suppressive activity (a) and proliferation index (b) of purified Treg cells co-cultured with CellTrace™ stained Tcon cells was measured at day 5. Data represent the mean ± s.d. ($n = 5$). The data were analyzed by one-way two-sided ANOVA.

3. Is the stabilization of PDL-1 on CD8 and Treg cells favoring immune exhaustion when a prolonged binding is achieved?

Response:

Thanks for the reviewer’s comment. The intact interaction between PD1 and its ligand PD-L1 could inhibit T cell activation and lead to their exhaustion or apoptosis, which lessened severity in many autoimmune diseases^{1,2,3}. Recent researches have demonstrated that the prolonged retention of the inhibitory immune checkpoint ligands on the pancreatic β cell membranes could improve their ability to energize the autoreactive T cells, and consequently increase their ability to reverse newly-onset diabetes⁴. In our work, we found that prolonged retention of PD-L1 on pancreatic β cell membranes through the dual-anchor coupling strategy reduced the IFN- γ and GzmB frequency in CD8⁺ T cells and enhanced the population of Tregs compared to the DBCO group both *in vitro* and *in vivo*. Moreover, we also demonstrated that the prolonged retention of PD-L1 reversed newly-onset diabetes and prolonged their survival. Thus, we believe that the stabilization of PD-L1 on pancreatic β cells could

favor immune exhaustion when a prolonged binding is achieved. We have also emphasized the important role of prolonged binding on immune suppression in our revised manuscript.

References:

1. Zhao, P. *et al.* Depletion of PD-1-positive cells ameliorates autoimmune disease. *Nat. Biomed. Eng.* **3**, 292-305 (2019).
2. Sugiura, D. *et al.* PD-1 agonism by anti-CD80 inhibits T cell activation and alleviates autoimmunity. *Nat. Immunol.* **23**, 399-410 (2022).
3. Zhang, X. *et al.* Engineered PD-L1-expressing platelets reverse new-onset type 1 diabetes. *Adv. Mater.* **32**, 1907692 (2020).
4. Au, K. M., Medik, Y., Ke, Q., Tisch, R. & Wang, A. Z. Immune Checkpoint-Bioengineered Beta Cell Vaccine Reverses Early - Onset Type 1 Diabetes. *Adv. Mater.* **33**, 2101253 (2021).

4. *Fig. 3B: there is a high fluorescence in the liver of mice injected with both H-NP and GLP1R-H-NP. Was the liver of these mice examined for histology, T cell infiltration and biochemical markers of inflammation (i.e., AST, ALT)?*

Response:

Thanks so much for the reviewer's suggestion. The liver of mice was examined for histology, T cell infiltration, and biochemical markers of inflammation. As shown in Fig. L3, negligible difference was observed in the liver of mice injected with H-NP and GLP1R-H-NP compared to healthy mice. We have also included the information in our revised manuscript and supporting information file.

Fig. L3. Histological analysis of liver. **a**, Representative H&E staining of liver sections from NOD mice after different treatments. Scale bar: 100 μ m. **b**, Representative anti-CD8 (red) stained liver sections from NOD mice after different treatments. Scale bar: 100 μ m. **c**, Blood biochemistry analysis of NOD mice after different treatments. AST: aspartate transferase, ALT: alanine transferase. Data are presented as mean \pm s.d. ($n = 4$)

5. Fig. 3D: A large variability in blood glucose levels is noted in mice undergoing DBCO/DSPE-PDL-1 treatment.

Response:

The blood glucose levels could fluctuate during the pathogenesis of type 1 diabetes, especially after the intervention of immunological strategies, which was consistent with previous reports^{1,2}. Thus, we believe the variability in blood glucose levels was a normal phenomenon.

1. Ben Nasr, M. *et al.* PD-L1 genetic overexpression or pharmacological restoration in hematopoietic stem and progenitor cells reverses autoimmune diabetes. *Sci. Transl. Med.* **9**, eaam7543 (2017).
2. Akbarpour M. *et al.* Insulin B chain 9-23 gene transfer to hepatocytes protects from type 1 diabetes by inducing Ag-specific FoxP3⁺ Tregs. *Sci. Transl. Med.* **7**, 289ra81-289ra81 (2015).

6. Fig. 3H: Increase in the number of islets cannot be clearly appreciated in the DBCO/DSPE-PDL-1 treated mice. The Authors should consider including a more representative, convincing picture.

Response:

We appreciate the reviewer's comment. We included a more representative H&E staining image (Fig. L4) in our revised manuscript.

Fig. L4. Representative H&E staining of pancreas sections from NOD mice after different treatments. Circles indicate islets in the pancreas. Black arrows indicate the infiltration of immune cells in pancreatic islets. Scale bar: 200 μ m.

7. Fig. 4: Changes in CD8 T cell phenotype (i.e., significantly lower IFN γ , granzyme B and cytokine levels) are noted when comparing G3 or G2 with G5. However, no significant differences are noted between G4 and G5.

Response:

Thanks so much for the reviewer's question. Even though both CD8⁺IFN- γ ⁺ and CD8⁺GzmB⁺ T cells were reduced in G5 compared to G4, no significant difference was obtained according to the statistical analysis perhaps due to the fact that DBCO-PD-L1 labeling could also protect pancreas from immune attack in a short time. We hypothesized that prolonged cultivation might amplify the differences between G4 and G5. Thus, we collected the pancreas from NOD mice on Day 10 post different treatments and performed flow cytometry analysis. More CD8⁺IFN- γ ⁺ and CD8⁺GzmB⁺ T cells were observed in G5 as compared to G4 ($P = 0.0352$ and $P = 0.0364$, respectively), which was attributed to the extended retention of PD-L1 on endogenous pancreatic β cell. We included the information in our revised manuscript.

Fig. L5. Characterizations of the T-cell status in the pancreas of diabetic NOD mice. **a**, Gating strategies for T cells analysis by flow cytometry in the pancreas on day 10 post-treatments. **b,c**, Representative plots of pancreas-infiltrating CD4⁺ and CD8⁺ T cells (**b**) and quantification of pancreas-infiltrating CD8⁺ T cells (**c**). **d,e**, Representative plots (**d**) and quantifications (**e**) of pancreas-infiltrating CD8⁺INF- γ ⁺ T cells in different treatment groups analyzed by the flow cytometry. **f,g**, Representative plots (**f**) and quantifications (**g**) of pancreas-infiltrating CD8⁺Gzmb⁺ T cells in different treatment groups analyzed by the flow cytometry. Data represent the mean \pm s.d. ($n = 5$). The data were analyzed by one-way two-sided ANOVA.

8. *Supplementary Fig. 16: Have the Authors measured cytokine producing cells among CD4 lymphocytes? Is the beneficial effect in the DBCO/DSPE-PDL-1 group linked to increased Tregs or decreased CD8 effectors?*

Response:

More CD4⁺FoxP3⁺ T cells were observed in the DBCO/DSPE-PDL-1 group compared to other groups (Supplementary Figure 23 and 24). The increased proportion of Tregs and decreased proportion of CD8 T cells contributed to the reversion of early-onset type 1 diabetes, which was also consistent with previous reports^{1,2}.

Reference:

1. Bluestone J. A., *et al.* Type 1 diabetes immunotherapy using polyclonal regulatory T cells. *Sci. Transl. Med.*, **7**, 315ra189-315ra189 (2015).
2. Yang S. J., *et al.* Pancreatic islet-specific engineered Tregs exhibit robust antigen-specific and bystander immune suppression in type 1 diabetes models. *Sci. Transl. Med.*, **14**, eabn1716 (2022).

9. Are systemic (*i.e.*, peripheral blood/spleen) and local (*i.e.*, pancreatic lymph nodes) compartments impacted in DBCO-PDL-1 and DBCO/DSPE-PDL-1 groups?

Response:

In order to explore whether the systemic and local compartments were impacted in DBCO-PD-L1 and DBCO/DSPE-PD-L1 groups, we collected the spleen and pancreatic lymph nodes and performed flow cytometry to evaluate the proportion of immune cells. As shown in Fig. L6, with the introduction of DBCO/DSPE-PD-L1, the proportion of CD8⁺ T cells in the pancreatic lymph nodes significantly reduced compared to the other groups. Previous report demonstrated that β -cell-specific CD8 T cells in NOD mice were primed in the pancreatic draining lymph node before they infiltrated the pancreas¹. Thus, we believe that the onset of type 1 diabetes contributed to the elevation of CD8⁺ cytotoxic T cells in the pancreatic lymph nodes, while the prolonged stabilization of PD-L1 on the pancreatic β cells could not only reverse the infiltration of CD8⁺ T cells, but restrain the activity of autoreactive T cells in the pancreatic lymph nodes. Moreover, the proportion of granzyme B (GzmB) positive CD8⁺ T cells and IFN- γ positive CD8⁺ T cells were also reduced after the administration of DBCO/DSPE-PD-L1. The DBCO/DSPE-PD-L1 treated NOD mice enhanced the population of Tregs as evidenced by the increased expression of FoxP3. We further investigated the status of T cells in the spleen of NOD mice. A lower proportion of CD8⁺ T cells was observed in the DBCO/DSPE-PD-L1 when compared to the untreated group and PD-L1 treated group (Fig. L7). Insignificant difference was observed in regard to GzmB positive CD8⁺ T cells and IFN- γ positive CD8⁺ T cells, perhaps due to the fact that locally modulation of immune microenvironment could not significantly affect the systemic immune systems. Now we included the discussion and results in our revised manuscript and supporting information files.

Reference:

1. Gearty S. V., *et al.* An autoimmune stem-like CD8 T cell population drives type 1 diabetes. *Nature*, **602**, 156-161 (2022).

Fig. L6. Characterizations of the T-cell status in the lymph node (LN) of diabetic NOD mice on day 10 post-treatments. **a,b**, Representative plots of LN-infiltrating CD4⁺ and CD8⁺ T cells (**a**) and quantification of LN-infiltrating CD8⁺ T cells (**b**). **c,d**, Representative plots (**c**) and quantifications (**d**) of LN-infiltrating CD8⁺INF- γ ⁺ T cells in different treatment groups analyzed by the flow cytometry. **e,f**, Representative plots (**e**) and quantifications (**f**) of LN-infiltrating CD8⁺GzmB⁺ T cells in different treatment groups analyzed by the flow cytometry. **g,h**, Representative plots (**g**) and quantification (**h**) of LN-infiltrating FoxP3⁺ T cells in different treatment groups analyzed by flow cytometry. Data represent the mean \pm s.d. ($n = 5$). The data were analyzed by one-way two-sided ANOVA.

Fig. L7. Characterizations of the T-cell status in the spleen of diabetic NOD mice on day 10 post-treatments. **a,b**, Representative plots of spleen-infiltrating CD4⁺ and CD8⁺ T cells (**a**) and quantification of spleen-infiltrating CD8⁺ T cells (**b**). **c,d**, Representative plots (**c**) and quantifications (**d**) of spleen-infiltrating CD8⁺INF- γ ⁺ T cells in different treatment groups analyzed by the flow cytometry. **e,f**, Representative plots (**e**) and quantifications (**f**) of spleen-infiltrating CD8⁺GzmB⁺ T cells in different treatment groups analyzed by the flow cytometry. **g,h**, Representative plots (**g**) and quantification (**h**) of spleen-infiltrating FoxP3⁺ T cells in different treatment groups analyzed by flow cytometry. Data represent the mean \pm s.d. ($n = 5$). The data were analyzed by one-way two-sided ANOVA.

10. Supplementary Fig. 17: *Foxp3* staining is almost unnoticeable. The Authors should provide a more representative picture.

Response:

Thanks for the reviewer's suggestion. We included a more representative image (Fig. L8) in our revised Supporting Information file.

Fig. L8. Characterization of Treg cells in the pancreas of NOD mice. Representative anti-CD4 (green)/anti-FoxP3 (red) dual-stained pancreas sections from NOD mice on day 5 post-treatments. Scale bar: 100 μ m.

11. Fig. 5D: The cumulative data presented do not adequately reflect the representative image, when considering lung, kidney, and spleen.

Response:

We have calculated the fluorescence intensity and corrected the cumulative data (Fig. L9) in our revised manuscript.

Fig. L9. Quantitative analysis of fluorescence intensity in different organs.

12. Fig. 5G: The Authors should check the data represented in this graph because G5 appears to have the highest paw thickness increase. The same applies to Supplementary Fig. 22B.

Response:

Thanks so much for the reviewer's carefulness. We have corrected the results (Fig. L10 and L11) in our revised manuscript and supporting information files.

Fig. L10. Right hind ankle joint swelling behaviors of arthritic mice. Data represent the mean \pm s.d. ($n = 5$). The data were analyzed by one-way two-sided ANOVA.

Fig. L11. *In situ* immobilization of PD-L1 ameliorates left joint swelling. **a**, Representative images of the left hind paw of mice in different treatment groups at the time of each treatment. **b**, Left hind ankle joint swelling behaviors of arthritic mice. Data represent the mean \pm s.d. ($n = 5$). The data were analyzed by one-way two-sided ANOVA.

13. As in the T1D model, are the beneficial effects noted in G5 linked to decreased effectors or increased Tregs?

Response:

Type 1 diabetes is a chronic autoimmune disease that results from autoreactive T cells destroying the insulin-producing pancreatic β cells. Previous reports have demonstrated that the recruitment of infiltrating Tregs and reduction of autoreactive T cells could enhance immunotolerance and reverse early-onset hyperglycemia^{1,2,3}. Thus, the decreased effectors or increased Tregs contributed to the beneficial effects noted in G5. We have also included the discussion in our revised manuscript.

Reference:

1. Bluestone J. A., *et al.* Type 1 diabetes immunotherapy using polyclonal regulatory T cells. *Sci. Transl. Med.*, **7**, 315ra189-315ra189 (2015).
2. Yang S. J., *et al.* Pancreatic islet-specific engineered Tregs exhibit robust antigen-specific and bystander immune suppression in type 1 diabetes models. *Sci. Transl. Med.*, **14**, eabn1716 (2022).
3. Bluestone J. A., *et al.* Immunotherapy: building a bridge to a cure for type 1 diabetes. *Science*, **373**, 510-516 (2021).

14. *Supplementary Fig. 29-30: There are rather high AST and ALT levels despite normal appearing liver histology. How do the Authors explain these findings?*

Response:

Thanks so much for the reviewer's question. In normal mice, the level of ALT, AST and UREA/BUN in plasma is ALT: 25-60 U/L; AST: 50-100 U/L¹. However, DBA/1J mice usually shows higher levels of ALS and AST, especially after the introduction of rheumatoid arthritis^{2,3}. We further replicated the experiment and measured the AST and ALT levels (Fig. L12). We included the data in our revised supporting information file.

Reference:

1. Sher Y. P., *et al.* Blood AST, ALT and urea/BUN level analysis. *Bio-protocol*, **3**, e931-e931 (2013).
2. Jung E. G., *et al.* Brazilin isolated from *Caesalpinia sappan* L. inhibits rheumatoid arthritis activity in a type-II collagen induced arthritis mouse model. *BMC Complem. Altern. M.*, **15**, 1-11 (2015).
3. Choi E. M., *et al.* A preliminary study of the effects of an extract of *Ligularia fischeri* leaves on type II collagen-induced arthritis in DBA/1J mice. *Food Chem. Toxicol.*, **46**, 375-379 (2008).

Fig. L12. Blood biochemistry analysis of DBA/1J mice after different treatments. AST: aspartate transferase, ALT: alanine transferase. Data are presented as mean \pm s.d. ($n = 4$)

15. Discussion: The Authors could elaborate on the advantages of the proposed approach over immunotherapies that are currently used or are in development. Limitations of the proposed approach should be mentioned.

Response:

Thanks so much for the reviewer's suggestion. We have added more discussion on the advantages of the proposed approach over immunotherapies that are currently used and the limitations of the proposed approach in our revised manuscript.

Minor points

16. Line 89: 'indistinctive' should be replaced with 'negligible'.

Response:

We have modified the description in our revised manuscript.

17. Line 252: please replace 'insignificant' with 'non-significant'.

Response:

We have modified the description in our revised manuscript.

Reviewer #2:

The manuscript of Wang et al. presents an approach aimed at increasing the density of expression of a checkpoint inhibitor ligand namely, the PD-L1 molecule within sites that are the target of an autoimmune reaction. In autoimmune diabetes, one of models analyzed the strategy should help counteracting the pathogenic capacity of CD4⁺ and CD8⁺ autoreactive T lymphocytes. The technical approach to increase the expression of the target molecule in a given cell is based on metabolic glycoengineering and bioorthogonal click chemistry to “anchor” the selected checkpoint inhibitor.

MAJOR COMMENTS

1) The strategy presented is not innovative per se; what is novel is the cell type the authors chose to target using the glycoengineered PD-L1 that is the insulin-producing beta-cell within the pancreas of the host. Not being a biochemist myself, I cannot judge on the method used to ensure that the coupling of the chosen molecule (that is in fact a modified form of PD-L1 in order to extend its half-life on beta-cells of the islets of Langerhans) was successful. I will therefore limit myself to reviewing the in vitro and in vivo biological data reported by the authors and their discussion on the importance of their findings in the context of the state of the art.

*To that aim, it is important to consider and confront the model and the results presented in this manuscript to the recent results published by the group of R. Tisch in *Advanced Materials*, 2021 (this paper is in fact referenced in the present manuscript as #13 (Au et al.)). This other group described the bioengineering (glycoengineering and bioorthogonal click chemistry) of a beta-cell line NIT-1 (established from NOD islets) to achieve increased surface expression of a combination of 3 checkpoint inhibitor ligands: PDL1, CD86 and Galectin69. The modified cell lines were placed into an acellular scaffold which was implanted subcutaneously in overtly diabetic NOD mice. The device could reproduce “a pancreas microenvironment” where diabetogenic T lymphocytes migrated and were, according to the authors “neutralized” via exhaustion, thereby reversing recent onset type 1 diabetes for about 2 months.*

Here the authors take a step further by proposing to couple the modified PD-L1 molecule directly to the beta-cells of the host using as a specific anchor molecule a ligand of the glucagon-like peptide-1 receptor (GLP1R). The therapeutic compound may therefore be administered quite simply by the i.v. route. All this to underscore that the innovation in this manuscript concerns the intrinsic characteristics of the therapeutic molecule: -which contains only PD-L1 (instead of the 3 molecules PD-L1, CD86 and Galectin 9 in the paper by Au et al) and -which includes a molecule allowing the direct targeting of beta-cells of the host, hence its simple administration. It is therefore necessary to closely analyze whether the efficacy criteria of the therapeutic compound are really significant. In vivo data upon administration in diabetic NOD mice (Figures 3d and e) would argue for a significant effect

quite similar to that in the manuscript of Au *et al.* The major problem is however that the cellular, histological data and, above all, confocal microscopy data do not show sufficiently clear images to conclude: - neither on the precise localization of the therapeutic product in the pancreas, - nor on the images characterizing the lymphocyte infiltrate in the pancreas, -nor on images of insulinitis in conventional histology. In particular, Fig. 3c, Fig 3i, Supplementary Fig 11a, Supplementary Fig 15, Supplementary Fig 17 should be revisited. Also, in Fig 2f the “colocalization of Min 6 cells or PD-L1 conjugated Min 6 cells and lymph node T cells were displayed by immunofluorescence staining” is really not easy to see; PD-1 red spots do not look like lymphocytes with a membrane, a cytoplasm, a nucleus. Higher magnifications and clear superimposed images are needed.

Response:

Thanks so much for the reviewer’s valuable comments. Click chemistry and bioorthogonal reactions provide powerful tools to label cells or monitor biological processes without disrupting biochemical processes, which offer significant potential for cellular manipulation. Such an approach has been tested in clinical trials of cancer therapy, while long-term cell engineering for assuaging autoimmunity remains challenging due to glycan/membrane recycling. In previous report, Au *et al.* also used this technology to modify NIT-1 cells with checkpoint inhibitor ligands, while the limited modulation duration due to glycan/membrane recycling significantly compromises their therapeutic outcomes. Although they used multivalent dendrimer anchor to provide a more effective conjugation and prolong the retention of the conjugated co-inhibitory checkpoint molecule on the NIT-1 cell surface, the immobilization still only lasted for less than 7 days, and only 37.5% of NOD mice remained normoglycemia after 100 days.

In our work, we have not only used bioorthogonal chemistry to conjugate checkpoint inhibitor ligands on the cell membrane, but also utilized physical insertion to prolong their immobilized duration for long-term alleviation of autoimmune diseases. Our double-anchor coupling strategy based on the integration of bioorthogonal chemistry and physical insertion is able to prevent the undesired loss of the conjugated proteins caused by the metabolic activity of cells, thus significantly enhancing the retention of the PD-L1 proteins on the target cells. In our approach, we demonstrated that over 45.9% of PD-L1 could be immobilized on the cell membrane of Min 6 cells after two-week incubation. In addition, *in vivo* experiments showed that PD-L1 could be specifically immobilized on the pancreatic β cells and persisted for at least 3 weeks, which was dramatically longer than previous reports. The prolonged retention of PD-L1 protected pancreatic β cells from immune attack and reversed the early-onset type 1 diabetes (57.1% normoglycemia after 100 days) and promoted the survival of NOD mice compared to the bioorthogonal chemistry alone method (14.3% normoglycemia after 100 days).

Moreover, such approach could be extended to the immobilization of multiple types of biomacromolecules. To the best of our knowledge, the described work is the first time to demonstrate a dual-anchor coupling approach to immobilize therapeutic biomacromolecules on endogenous cells both *in vitro* and *in vivo*, based on bioorthogonal reactions and membrane insertion.

According to the reviewer's valuable comments, we have carefully revised and improved the manuscript by supplementing new experimental data (Fig. L13-L20), rephrasing the manuscript and including key references.

Fig. L13. Fluorescent imaging of the pancreas from NOD mice *i.v.* injected with Cy5-labeled PD-L1 analogs on day 7. Scale bar: 50 μm .

Fig. L14. Representative anti-insulin (red)/anti-glucagon (green) dual stained pancreas sections from NOD mice on day 5 post-treatments. Scale bar: 200 μm .

Fig. L15. The dual-anchor coupling strategy prolonged the immobilization of PD-L1 on islets. **a,b**, Fluorescent imaging (**a**) and quantifications (**b**) of fluorescent signals of the pancreas from NOD

mice *i.v.* injected with Cy5-labeled PD-L1 analogs at different time points (Green: Glucagon; Red: Cy5-labeled PD-L1 analogs). Scale bar: 50 μ m. Data represent the mean \pm s.d. ($n = 6$).

Fig. L16. Characterization of CD4⁺ T cells in the pancreas of NOD mice. Representative anti-CD4 (red)/anti-glucagon (green) dual stained pancreas sections from NOD mice on day 5 post-treatments. Scale bar: 100 μ m.

Fig. L17. Characterization of CD8⁺ T cells in the pancreas of NOD mice. Representative anti-CD8 (red)/anti-glucagon (green) dual stained pancreas sections from NOD mice on day 5 post-treatments. Scale bar: 200 μ m.

Fig. L18. Characterization of Treg cells in the pancreas of NOD mice. Representative anti-CD4 (green)/anti-FoxP3 (red) dual-stained pancreas sections from NOD mice on day 5 post-treatments. Scale bar: 100 μ m.

Fig. L19. Colocalization of Min 6 cells or PD-L1 conjugated Min 6 cells and lymph node T cells were displayed by immunofluorescence staining. Scale bar: 5 μ m.

Fig. L20. The dual-anchor coupling strategy reduced the incidence of insulinitis. **a**, Representative H&E staining of pancreas sections from NOD mice after different treatments. Circles indicate islets in the pancreas. Black arrows indicate the infiltration of immune cells in pancreatic islets. Scale bar: 200 μ m. **b**, Insulinitis scores of pancreatic sections. Data represent the mean \pm s.d. ($n = 5$).

2) I am puzzled by control data in figure 3g: how could the authors maintain untreated diabetic NOD mice for a median time of about 60 days. In colonies with a reasonable diabetes incidence >60% in females once diabetic animals die or are culled within 2 weeks since they lose weight and waste away very quickly. What is the incidence of diabetes in the NOD females used?

Response:

Thanks so much for the reviewer's question. The NOD mice in our experiment were purchased from Beijing HFK Bioscience Co., LTD. The incidence of diabetes is around 50~60%. We conducted the experiments when the blood glucose levels were above 250 mg dL⁻¹ for two consecutive days. After the incidence of diabetes, the blood glucose levels of NOD mice started rising, which usually took days to weeks until the blood glucose levels were above 600 mg dL⁻¹ (Figure 3d). Once the mice become such hyperglycemia, they will die or be culled soon since they lose weight and waste away very quickly. In order to confirm the results, we replicated the experiments by increasing the sample numbers. As shown in Fig. L21, over 50 % of NOD mice without any treatment could live for more than 40 days, which was consistent with previous reports^{1,2,3}. We have included the data in our revised manuscript.

Fig. L21. The dual-anchor coupling strategy prolonged the survival of NOD mice. Survival of the mice with different treatments. ($n = 20$).

Reference:

1. Gao S., *et al.* Tetrahedral framework nucleic acids induce immune tolerance and prevent the onset of type 1 diabetes. *Nano Letters*, **21**, 4437-4446 (2021).
 2. Ben N. M., *et al.* PD-L1 genetic overexpression or pharmacological restoration in hematopoietic stem and progenitor cells reverses autoimmune diabetes. *Sci. Transl. Med.* **9**, eaam7543 (2017).
 3. Zhang, X. *et al.* Engineered PD-L1-expressing platelets reverse new - onset type 1 diabetes. *Adv. Mater.* **32**, 1907692 (2020).
 4. Zhao, P. *et al.* Depletion of PD-1-positive cells ameliorates autoimmune disease. *Nat. Biomed. Eng.* **3**, 292-305 (2019).
- 3) *Data in collagen-induced arthritis appear preliminary. The authors should concentrate on autoimmune diabetes.*

Response:

Thanks so much for the reviewer's suggestion. We included the study in the collagen-induced arthritis model to indicate our approach is not disease specific. We have supplemented new experimental data, rephrased the manuscript and included key references to improve the manuscript.

Reviewer #3:

In this manuscript by Wang et al., the authors describe a novel technique for anchoring exogenous PD-L1 on the membrane of target cells within in mouse, treating autoimmunity in two disease models. These are intriguing data with convincing in vivo results. However, the correlative data do not sufficiently convince me that this technique is working through the mechanism that the author's propose. I have the following concerns.

General concerns regarding impact

1. The authors show that their technique relatively specifically localizes PD-L1 to the pancreas. Their data are not convincing that they can do the same for the joints. The authors have demonstrated some proof of principle in one model, but without demonstrating more specificity for their labeling process, I am not convinced that this technique will necessarily be an improvement over broad immunosuppression. This does not necessarily negate the value of their findings, but it does warrant discussion in the text as a limitation.

Response:

Thanks so much for the reviewer's valuable suggestion. Previous reports have demonstrated that a ligand of the glucagon-like peptide-1 receptor (GLP1R) could efficiently target pancreatic β cells^{1,2}. In the T1D model, GLP1R-H-NPs loaded with Ac₄ManNAz were injected into NOD mice through tail-vein, and then β cells processed azido sialic acid derivatives on their membrane *via* metabolic glycoengineering. Afterward, DBCO-PD-L1 or DBCO/DSPE-PD-L1 was administered respectively. By selectively conjugating to the azido groups on the β cell membrane, PD-L1 could be anchored on the cell membrane through click chemistry. In the rheumatoid arthritis model, the Ac₄ManNAz-loaded H-NPs were intraarticularly injected in the ankle joints of DBA/1 mice. Therefore, chondrocytes on the surface of articular cartilage could express azido sialic acid derivatives on their membrane. PD-L1 analogs could be anchored on the cell membrane through click chemistry. Such an approach was inspired by previous reports for cancer therapy³. By changing the targeting moieties or *in situ* injection, our labeling approach could be expanded to various autoimmune diseases. We have added the relevant discussion in the revision.

Reference:

1. Ämmälä, C. *et al.* Targeted delivery of antisense oligonucleotides to pancreatic β -cells. *Sci. Adv.* **4**, eaat3386 (2018).

2. Knerr L., *et al.* Glucagon like peptide 1 receptor agonists for targeted delivery of antisense oligonucleotides to pancreatic beta cell. *J. Am. Chem. Soc.*, **143**, 3416-3429 (2021).
3. Wang H., *et al.* Selective *in vivo* metabolic cell-labeling-mediated cancer targeting. *Nat. Chem. Biol.*, **13**, 415-424 (2017).

2. *From the data as presented, it is not clear how long the PD-L1 conjugate will remain in place and how long the immune suppressing effects will last. The impact would be greatly enhanced were much later time points (4 months, 6 months) examined, particularly for persistence of the exogenous PD-L1. Again, this does not necessarily mean that the current manuscript is insufficient, but it is a limitation of the data as provided.*

Response:

Thanks so much for the reviewer's suggestion. The prolonged retention of PD-L1 on target cells has been evaluated *in vitro* (Figure 2b-d, Supplementary Figure 7 and 8). The PD-L1 conjugate could remain on the cell membrane for at least 2 weeks, which was significantly longer than previous reports (less than 1 week)¹. In our approach, we demonstrated that over 45.9% of PD-L1 could be immobilized on the cell membrane of Min 6 cells after two-week incubation, indicating that the dual-anchor coupling approach could significantly prolong the immobilization of immune checkpoints on the target cells. In addition, *in vivo* experiments showed that PD-L1 could be specifically immobilized on the pancreatic β cells and persisted for at least 3 weeks, which was dramatically longer than previous reports (Fig. L22). We are also performing more experiments to investigate the long-term conjugation duration as well as immune suppressing effects, and would incorporate relevant results in the follow-up publications.

Reference:

1. Au, K. M., Medik, Y., Ke, Q., Tisch, R. & Wang, A. Z. Immune Checkpoint - Bioengineered Beta Cell Vaccine Reverses Early - Onset Type 1 Diabetes. *Adv. Mater.* **33**, 2101253 (2021).

Fig. L22. The dual-anchor coupling strategy prolonged the immobilization of PD-L1 on islets. a,b, Fluorescent imaging (a) and quantifications (b) of fluorescent signals of the pancreas from NOD

mice *i.v.* injected with Cy5-labeled PD-L1 analogs at different time points. Scale bar: 50 μm . Data represent the mean \pm s.d. ($n = 6$).

Concerns with the Results

3. Although using FITC conjugated DBCO/DSPE-PD-L1 demonstrates that the construct is associated with the target cells of interest, it is important to determine whether this PD-L1 is appropriately folded. Using fluorophore conjugated PD-1 as a probe as well as a fluorophore conjugated anti-PD-L1 antibody would both be valuable to assess surface expression of folded PD-L1 capable of associating with PD-1.

Response:

Thanks so much for the reviewer's suggestion. We evaluated whether the structure of PD-L1 conjugates was appropriately folded utilizing a fluorophore-conjugated anti-PD-L1 antibody (a-PD-L1). Through confocal microscope imaging, we found that more anti-PD-L1 antibody was localized on DBCO/DSPE-PD-L1-labeled cell membrane compared to the cell alone group (Fig. L23). In addition, we also evaluated the functional activities of PD-L1 conjugates through flow cytometry. As shown in Figure 2g and h, the introduction of anti-PD-L1 antibody could reduce the protective capacity of PD-L1 conjugates towards Min-6 cells. Thus, we believe that the structure of PD-L1 conjugates was appropriately folded. Now we included the discussion in our revised manuscript.

Fig. L23. The dual-anchor coupling strategy enhanced the expression of PD-L1 on Min 6 cells. a,b, Confocal imaging (a) and quantitative assay (b) of Min 6 cells stained with a-PD-L1 (red). Scale

bar: 50 μm . Data represent the mean \pm s.d. ($n = 6$). The data were analyzed by two-tailed Student's t-test.

4. Figure 1d looks like DBCO/DSPE-PD-L1 loses fluorescence faster than DBCO-PD-L1, perhaps I am missing something, but this appears to be the opposite of the author's claim. I suspect that this is a typo.

Response:

Thanks so much for pointing this out. We have corrected the labeling in our revised manuscript. We have also included the statistical analysis from flow cytometry assay in our revised Supporting Information.

Fig. L24. The dual-anchor coupling strategy prolonged the immobilization of PD-L1 on the cell membrane. **a** Flow cytometry of Min 6 cells labelled with FITC-PD-L1 analogs at predetermined time points. **b** Quantitative assay of Min 6 cells labeled with PD-L1 analogs on Day 3 *via* flow cytometry. Data represent the mean \pm s.d. ($n = 3$). The data were analyzed by two-tailed Student's t-test.

5. The "G" labeling on this figure is confusing. I recommend having a clear legend with the treatment groups visible near the bar graphs. Color coding is sufficient to identify the groups if a legend is present.

Response:

Thanks so much for the reviewer's suggestion. We have added clear legends with the treatment groups in our revision.

6. I would like anti-PD-1 treated cultures or PD-1 KO T cells as an additional control for figure 2 g and h. It would also be valuable to have a negative control cell line as a target that you would not expect these T cells to activate in the presence of. This cell line with or without DBCO/DSPE-PD-L1 could be exposed to T cells; additional controls would not be necessary provided that these two conditions were negative.

Response:

Thanks so much for the reviewer’s suggestion. We pretreated T cells with anti-PD-1 and incubated these T cells with Min 6 cells and DBCO/DSPE-PD-L1-labeled Min 6 cells, respectively. Negligible difference could be observed, indicating that the interaction between PD-1 and PD-L1 contributed to the T cell exhaustion (Fig. L25).

Fig. L25. Characterizations of the status of anti-PD-1 treated T cells in different treatment groups analyzed by flow cytometry. a, Schematic illustration revealed that the introduction of anti-PD-1 blocked the interactions between T cells and Min 6 cells. **b,c,** Representative plots (a) and

quantification (**b**) of CD8⁺ T cells in different treatment groups analyzed by the flow cytometry. **d,e**, Representative plots (**d**) and quantifications (**e**) of CD8⁺INF- γ ⁺ T cells in different treatment groups analyzed by the flow cytometry. **f,g**, Representative plots (**f**) and quantifications (**g**) of CD8⁺GzmB⁺ T cells in different treatment groups analyzed by the flow cytometry. **h,i**, Representative plots (**h**) and quantification (**i**) of CD4⁺FoxP3⁺ T cells in different treatment groups analyzed by flow cytometry. Data represent the mean \pm s.d. ($n = 5$)

7. The data in 3h and 3i need to be quantified in the main figure. This would also be much more convincing if it contained PD-L1 immunofluorescence. PD-L1 is typically expressed in the NOD pancreas, so it would be important to demonstrate that the total PD-L1 level is actually increased by this treatment in order to confirm the proposed mechanism of action.

Response:

Thanks so much for the reviewer's suggestion. The quantifications of Figure 3h and 3i were performed in our revised manuscript. In addition, anti-PD-L1 immunofluorescent imaging was conducted. Higher PD-L1 levels were observed in the DBCO/DSPE-PD-L1 treated group compared to that in other groups, revealing that reversing of early-onset type 1 diabetes was attributed to the increased expression of PD-L1 on pancreatic β cells.

Fig. L26. PD-L1 bioengineering reverses the early-onset type 1 diabetes in the NOD mice. a, Insulinitis scores of pancreatic sections quantified from H&E staining. **b**, Quantification of insulin⁺ β -cells in the pancreas sections from Figure 3i. **c**, Quantification of the fluorescence intensities of PD-L1 in the pancreas sections from Figure 3j. Data represent the mean \pm s.d. ($n = 5$).

8. *The BSA conjugation seems much less specific overall than the GLP1R conjugate. Although the authors do not see toxicity from this, this is almost certainly because these mice were not undergoing an infectious challenge (something that will certainly not be the case with free living people). The authors should acknowledge this lack of specificity and provide more of a rationale for BSA conjugation than they currently do.*

Response:

Thanks so much for the reviewer's suggestion. In this work, we just used BSA as a model protein for PD-L1 to investigate the conjugating behavior *in vivo*. In the rheumatoid arthritis model, the Ac₄ManNAz-loaded H-NPs were intraarticularly injected in the ankle joints of DBA/1 mice. Then, we injected free Cy5-BSA, Cy5-DBCO-BSA, and Cy5-DBCO/DSPE-BSA to evaluate the bioengineering efficiency of this dual-anchor coupling approach. We have also provided a clear description in the revision.

9. *I think that figure 4g is mislabeled using the "G" labeling format. If not, then these data appear to show that the DBCO/DSPE-PD-L1 construct actually makes the disease worse?*

Response:

Thanks so much for pointing this out. The "G" labeling format was mislabeled and we have corrected the labeling in our revised manuscript.

General Minor Comments

10. *In general, the English needs some work throughout the manuscript. Several of the sentences have an odd structure that is difficult to follow.*

Response:

Thanks for the reviewer's kind suggestion. We have thoroughly polished the whole manuscript.

11. *The title is not accurate. The authors only used PD-L1, thus "Immune Checkpoints" is more general than they have shown.*

Response:

Thanks so much for the reviewer's suggestion. We have changed the title to "An *In Situ* Dual-Anchoring Strategy for Long-Term Immobilization of PD-L1 to Treat Autoimmune Diseases".

12. *“and inflammatory bowel disease developed from the imbalance of the immune systems,” this is an over simplification. We really don’t know why these diseases develop.*

Response:

We thank the reviewer’s comment. We have changed the description in the revision.

13. *“the various side effects and long-term medication have severely reduced the quality of life for patients” the quality of life impact of many immunosuppressants is relatively small. The main drawback is often that the medications are not sufficiently effective, particularly for a disease like Type 1 Diabetes, where they have no role.*

Response:

We have changed the information in our revised manuscript.

REVIEWER COMMENTS

Reviewer #1 (Remarks to the Author):

The Authors have answered most of my comments with the inclusion of additional experimental data.

I have only one minor point related to current Fig. 3H where the panel labelled as 'untreated' appears at a different magnification, when compared to others panels. Could the Authors clarify this?

Reviewer #2 (Remarks to the Author):

The authors have addressed satisfactorily the issues I raised in my review of the original manuscript.

I still think that the arthritis data is less documented/convincing than the autoimmune diabetes one. However, I leave it to the Editor to decide whether or not they are included in the final version of the manuscript.

Reviewer #3 (Remarks to the Author):

The authors have addressed some but not all of my concerns. My remaining concerns are as follows:

1. The authors show that their technique relatively specifically localizes PD-L1 to the pancreas. Their data are not convincing that they can do the same for the joints, and simply injecting all joints with this construct is not feasible nor is it an improvement upon current techniques (we can already inject immune suppressants into individual joints with a joint specific response). The authors therefore have demonstrated some proof of principle in one model, but really have not laid the groundwork to convince me that this is a general approach. Nevertheless, focusing this manuscript on the NOD mice would be a reasonable response.

2. The authors still show a very short duration of action. If this technique only lasts for a few weeks, then patients will require repeat dosing. They should demonstrate that that still works in an animal model (treating NOD mice over months), or they will have to limit the scope of their claims – this is a proof of principle for short term treatment, but may not lead to long term disease control.

3. Antibodies are not sufficient alone to demonstrate folding of a molecule. Binding of the ligand (PD-1) is greatly preferable.

4. In making my comment about PD-1 blockade (or KO) cells, I had presumed that the authors would include a positive control (WT or unbound cells). As it is, the current experiment cannot be distinguished from a technical failure of the conjugate.

Responses to reviewer's comments

We sincerely thank the reviewers for their valuable comments and suggestions. Below we have provided responses to the comments and have accordingly revised the manuscript.

Reviewer #1:

The Authors have answered most of my comments with the inclusion of additional experimental data.

I have only one minor point related to current Fig. 3H where the panel labelled as 'untreated' appears at a different magnification, when compared to others panels. Could the Authors clarify this?

Response:

We greatly appreciate the reviewer for the positive feedback. We corrected the image in our revised manuscript.

Fig. L1. Representative H&E staining of pancreas sections from NOD mice on day 5 post-treatments. Circles indicate islets in the pancreas. Black arrows indicate the infiltration of immune cells in pancreatic islets. Scale bar: 200 μ m.

Reviewer #2:

The authors have addressed satisfactorily the issues I raised in my review of the original manuscript. I still think that the arthritis data is less documented/convincing than the autoimmune diabetes one. However, I leave it to the Editor to decide whether or not they are included in the final version of the manuscript.

Response:

We really appreciate the reviewer's positive feedback. Based on the editorial comment, we still include the arthritis data in the final version.

Reviewer #3:

The authors have addressed some but not all of my concerns. My remaining concerns are as follows:

- 1. The authors show that their technique relatively specifically localizes PD-L1 to the pancreas. Their data are not convincing that they can do the same for the joints, and simply injecting all joints with this construct is not feasible nor is it an improvement upon current techniques (we can already inject immune suppressants into individual joints with a joint specific response). The authors therefore have demonstrated some proof of principle in one model, but really have not laid the groundwork to convince me that this is a general approach. Nevertheless, focusing this manuscript on the NOD mice would be a reasonable response.*

Response:

Thanks for the reviewer's comment. In this work, we aim to propose a new long-term immobilization methodology of biomacromolecules such as PD-L1 on target cells/organs for the amelioration of autoimmune disease. In the rheumatoid arthritis model, we labeled the chondrocytes with azido groups in the articular cartilage *via* an intra-articular injection of Ac₄ManNAz-loaded nanoparticles. Through bioorthogonal chemistry and physical insertion, the chondrocytes could persistently express azido groups for a long period of time. During this time, PD-L1 analogs (DBCO/DSPE- PD-L1) could be administered *via* tail-vein repeatedly. Our approach only needs a single injection of Ac₄ManNAz into the joints without multiple intra-articular injections of immune suppressants that have been used in the current techniques. Similar approaches have been demonstrated in the treatment of other diseases, such as cancer¹. In addition, several targeting moieties and strategies have been reported to achieve rheumatoid arthritis-targeted drug delivery²⁻⁴, which could also be utilized to deliver Ac₄ManNAz for the labeling of chondrocytes. In follow-up work, we will explore this technique for the treatment of other types of autoimmune diseases, such as autoimmune encephalomyelitis.

References:

1. Wang H., *et al.* Selective *in vivo* metabolic cell-labeling-mediated cancer targeting. *Nat. Chem. Biol.*, **13**, 415-424 (2017).
2. Chen J., *et al.* Photoacoustic image-guided biomimetic nanoparticles targeting rheumatoid arthritis. *Proc. Natl. Acad. Sci. U.S.A.*, **119**, e2213373119 (2022).
3. Lee S., *et al.* Targeted chemo-photothermal treatments of rheumatoid arthritis using gold half-shell multifunctional nanoparticles. *ACS Nano*, **7**, 50-57 (2013).
4. Syed A., *et al.* Potential of targeted drug delivery systems in treatment of rheumatoid arthritis. *J. Drug Deliv. Sci. Technol.*, **53**, 101217 (2019).

2. The authors still show a very short duration of action. If this technique only lasts for a few weeks, then patients will require repeat dosing. They should demonstrate that that still works in an animal model (treating NOD mice over months), or they will have to limit the scope of their claims – this is a proof of principle for short term treatment, but may not lead to long term disease control.

Response:

Thanks so much for the reviewer's comment. Due to cell glycan/membrane recycling, proteins or glycan on the cell membrane were not stable. Utilizing our dual-anchor coupling strategy, we significantly extended the immobilization period of PD-L1 on target cells/organs, which was longer than most of the current cell labeling approaches such as chemical reactions. Most importantly, even though our approach immobilized target cells/organs with PD-L1 for a few weeks, the hyperglycemia symptoms were significantly alleviated for more than three months. We hypothesized that the prolonged retention of PD-L1 protected pancreatic β cells from immune attack in early stage of type 1 diabetes (T1D). The remission of auto-reactive immune response could reverse the early-onset T1D, and such effects may last for quite a long time. In our experiment, more than 57.1% of early-onset T1D mice remained normoglycemia after 100 days. Thus, we believe our approach achieved a long-term alleviation of autoimmune diseases.

3. Antibodies are not sufficient alone to demonstrate folding of a molecule. Binding of the ligand (PD-1) is greatly preferable.

Response:

Thanks for the reviewer's comment. We evaluated whether the structure of PD-L1 conjugates was appropriately folded utilizing a fluorophore-conjugated PD-1. Through confocal microscope imaging, we found that more PD-1 was localized on DBCO/DSPE-PD-L1-labeled cell membrane compared to the cell alone group (Fig. L2). Now we have included the information in our revised supporting information file.

Fig. L2. The dual-anchor coupling strategy enhanced the expression of PD-L1 on Min 6 cells. a,b, Confocal imaging (a) and quantitative assay (b) of Min 6 cells stained with PD-1 (green). Scale bar: 50 μ m. Data represent the mean \pm s.d. ($n = 5$). The data were analyzed by two-tailed Student's t -test.

4. In making my comment about PD-1 blockade (or KO) cells, I had presumed that the authors would include a positive control (WT or unbound cells). As it is, the current experiment cannot be distinguished from a technical failure of the conjugate.

Response:

Thanks so much for the reviewer's suggestion. We pretreated T cells with anti-PD-1 and incubated these T cells with Min 6 cells, DBCO-labeled Min 6 cells, and DBCO/DSPE-PD-L1-labeled Min 6 cells, respectively. Unbound T cells were incubated with DBCO/DSPE-PD-L1-labeled Min 6 cells as a positive control. Negligible differences among the anti-PD-1 treated groups could be observed (Fig. L3). However, when the unbound T cells were incubated with DBCO/DSPE-PD-L1-labeled cells, the proportion of CD8⁺ T cells and the frequency of IFN- γ and GzmB production were remarkably reduced, indicating that the interaction between PD-1 and PD-L1 contributed to the T cell exhaustion.

Fig. L3. Characterizations of the status of anti-PD-1 treated T cells in different treatment groups analyzed by flow cytometry. **a**, Schematic illustration revealed that the introduction of anti-PD-1 blocked the interactions between T cells and Min 6 cells. **b,c**, Representative plots (**a**) and quantification (**b**) of CD8⁺ T cells in different treatment groups analyzed by flow cytometry. **d,e**, Representative plots (**d**) and quantifications (**e**) of CD8⁺IFN- γ ⁺ T cells in different treatment groups analyzed by the flow cytometry. **f,g**, Representative plots (**f**) and quantifications (**g**) of CD8⁺GzmB⁺ T cells in different treatment groups analyzed by flow cytometry. **h,i**, Representative plots (**h**) and quantification (**i**) of CD4⁺FoxP3⁺ T cells in different treatment groups analyzed by flow cytometry. Data represent the mean \pm s.d. ($n = 5$)

REVIEWERS' COMMENTS

Reviewer #3 (Remarks to the Author):

The new data the authors have shared does address my concerns for points 3 and 4 - I believe that the PD-L1 is folded and that their effect depends on interaction between their immobilized PD-L1 and PD-1 on the T cells. As a minor point, the labeling of their new figure (L3) is confusing. All of the groups received anti-PD-1 as far as a can tell, except for the "unbound" control. I think it would be easier to read the figure if (+anti-PD-1) were added to call the groups except for the "unbound group" which I think would be better positioned next to the negative control.

My concerns about the scope of the authors' claims remain the same. The authors have cleared demonstrated proof of principal for short term treatment in a diabetes model. The arthritis data are less convincing, and I think that model has so little resemblance to a human treatment situation that it does not sufficiently bolster their claim to have developed a general treatment for autoimmune disease.

Responses to reviewer's comments

We sincerely thank the reviewer for their valuable comments and suggestions. Below we have provided responses to the comments and have accordingly revised the manuscript.

Reviewer #3:

The new data the authors have shared does address my concerns for points 3 and 4 - I believe that the PD-L1 is folded and that their effect depends on interaction between their immobilized PD-L1 and PD-1 on the T cells. As a minor point, the labeling of their new figure (L3) is confusing. All of the groups received anti-PD-1 as far as a can tell, except for the "unbound" control. I think it would be easier to read the figure if (+anti-PD-1) were added to call the groups except for the "unbound group" which I think would be better positioned next to the negative control.

Response:

Thanks so much for the reviewer's valuable suggestion. We have changed the labeling in the revision.

Fig. L1. Characterizations of the status of anti-PD-1 treated T cells in different treatment groups analyzed by flow cytometry. **a**, Schematic illustration revealed that the introduction of anti-PD-1 blocked the interactions between T cells and Min 6 cells. **b,c**, Representative plots (**a**) and quantification (**b**) of CD8⁺ T cells in different treatment groups analyzed by the flow cytometry. **d,e**, Representative plots (**d**) and quantifications (**e**) of CD8⁺INF- γ ⁺ T cells in different treatment groups analyzed by the flow cytometry. **f,g**, Representative plots (**f**) and quantifications (**g**) of CD8⁺GzmB⁺ T cells in different treatment groups analyzed by the flow cytometry. **h,i**, Representative plots (**h**) and quantification (**i**) of CD4⁺FoxP3⁺ T cells in different treatment groups analyzed by flow cytometry. Data represent the mean \pm s.d. ($n = 5$)

My concerns about the scope of the authors' claims remain the same. The authors have cleared demonstrated proof of principal for short term treatment in a diabetes model. The arthritis data are less convincing, and I think that model has so little resemblance to a human treatment situation that it does not sufficiently bolster their claim to have developed a general treatment for autoimmune disease.

Response:

Thanks so much for the reviewer's comment. We have changed the title from "An *In Situ* Dual-Anchoring Strategy for **Long-Term** Immobilization of PD-L1 to Treat Autoimmune Diseases" to "An *In Situ* Dual-Anchoring Strategy for **Enhanced** Immobilization of PD-L1 to Treat Autoimmune Diseases" and also adjusted the relevant descriptions in the revision. More discussion about the potential limitations has also been added in the end of the Discussion section as follows: In addition, the extension of such an approach to the treatment of other autoimmune diseases that resembles a human treatment situation needs further exploration.